



**Weakening anomalies of East Asian Summer Precipitation Influenced by the Tibetan Plateau Warming Amplification**

Mei Liang[1,2], Jianjun Xu[1,2], Liguang Wu[3],XiangdeXu[4]

[1]South China Sea Institute of Marine Meteorology, Guangdong Ocean University, Zhanjiang, China.

[2] Guangdong Ocean University, Zhanjiang, China

[3] Department of Atmospheric and Ocean Science, Fudan University, Shanghai, China.

[4]State Key Laboratory of Disastrous Weather, China Academy of Meteorological Sciences, Beijing, 100081, China

Corresponding author address: Prof. Jianjun Xu

South China Sea Institute of Marine Meteorology

Guangdong Ocean University, Zhanjiang, Guangdong 524088, China

E-mail: gmuxujj@163.com

Mei Liang's ORCID ID: 0000-0002-2058-9372


**Abstract:** The present study documents the on East Asian precipitation in summer influenced by elevation-dependent temperature change over the Tibetan Plateau (TP). The temperature of the TP and

its surrounding areas decreases with altitude by 0.43–0.45°C/100 meters, which is lower than the 0.6°C/-100 meters in the troposphere. The magnitude of the trend of temperature increase with elevation and the amplification of warming over the TP comprise an important feature of the temperature change. TP warming is consistently the important contributor to the variation of East Asian precipitation in summer from 1979 to 2016, but their relationship weakens as the warming over the TP amplifies. The

southern flood–northern drought pattern is weak compared with when the TP warming trend has been removed. Warming amplification of the TP may weaken the atmospheric circulation anomaly pattern. The rate of anticyclonic circulation strengthening has slowed in the upper and lower levels over Mongolia in East Asia, which leads to the "northern drought" feature weaken. Meanwhile striking cyclonic circulation anomalies are reduced in the southeastern part of China, the northern part of the

South China Sea, and the northwestern Pacific. The atmospheric response to TP warming might be related to two distinct Rossby wave trains. After TP warming amplifying, one in the extratropics that moves along the upper-level westerly jetstream and the other in the tropics that moves along the low-level monsoon westerly have weakened. The downdrafts prevailed in the Northern Asia receded, which is conducive to precipitation in the area. Updrafts prevailed in the Southeast Asia is deteriorating,

which is not advantageous to precipitation.



## 1 Introduction

The Tibetan Plateau (TP) has an average elevation of more than 4,000 m, making it the largest and highest plateau terrain in the world. It exerts a profound influence on regional and even global climate

via its thermal and dynamic forcing mechanism(Ye et al.1957; Ye and Gao 1979; Yanai et al. 1992; Ye and Wu 1998; Duan and Wu 2005,2008; Duan et al. 2010,2012,2013; Wu et al. 2012a, 2012b, 2014; Yao et al. 2012). On the geological time scale, the water circulation around the plateau and the convective activity of the plateau interact with the Asian monsoon system to have a profound impact on the precipitation of the Asian monsoon(Xu et al. 2002, 2010, 2015; Lu et al., 2005).

There are significant differences between the characteristics of climate change in high-altitude and low-altitude regions (Beniston et al. 1997; Liu and Chen 2000;Beniston et al. 2003; Liu et al. 2012; Wang et al. 2014;Zhu et al. 2018) Therefore, under global warming, the characteristics of temperature changes in high-altitude regions and their impact on the surrounding environment and even the global climate constitute a hot topic for climate scientists (Liu and Chen, 2000; Seidel and Free, 2003;

Beniston, 2003; Liu et al., 2009).Liu et al (2009) divided 116 weather stations and regional model grids into elevation zones in intervals of 500meters to examine the relationship between climate warming and elevation. With corroborating datasets, they confirmed the elevation-dependency of monthly mean minimum temperature in and around the TP. The warming is more prominent at higher elevations than at lower elevations, especially during winter and spring, and such a tendency may continue in future

climate change scenarios. However, Wang et al. (2014) pointed out that the terrain amplification effect of the rate of temperature change in high-altitude areas is difficult to identify based on the globally averaged field, and shows significant regional characteristics. Many studies (Beniston et al.1997; Liu et al.2009; Rangwala and Miller 2012; Ohmura 2012) have found a terrain warming amplification effect on the TP, but it remains an uncertain phenomenon, largely because of the inadequacies of observations

at high altitude (Rangwala and Miller 2012; Ohmura 2012) and region-specific conditions (Beniston et al.1997; Liu et al.2009; Ohmura 2012). Therefore, there are several open questions worthy of further consideration. For example: How does the temperature of the TP and its surrounding area change with altitude? And do different data reflect the difference in the amplification of the terrain?

Observational studies have exhibited that the uplift of the TP has a profound influence on the

formation of the East Asian summer monsoon. Variations in TP heating also have a profound effect on the interannual and interdecadal variations of East Asian monsoon in the modern record(Ye et al. 1992; Ding 1992; Wu et al.1998, 2012; Lu et al. 2005; Liu et al. 2012; Xu et al. 2008; Boos et al. 2013). Wu


et al. (1998) pointed out that the TP affects the East Asian monsoon by inducing air pumping over the TP and producing cyclonic spiral zonal deviation in the lower troposphere and negative vorticity in the

upper troposphere over the plateau, which plays an important role in the outbreak and maintenance of the Asian monsoon. However, the physical processes by which a persistent warming over the TP affects downstream East Asian rainfall remain elusive. Xu et al.(1998, 2002) pointed out that the TP is the source of monsoon water vapor transport or the "transportation station" in the Yangtze River basin. The route is mainly from the east of the Philippines, passing through the South China Sea, the Bay of Bengal,

and the eastern part of the plateau to the Yangtze River basin. This route affects precipitation over China and even East Asia as a whole. By affecting the Qinghai–Tibet high, western Pacific subtropical high, Asian monsoon, and circulation in the middle and high latitudes of Europe and Asia, thermal anomalies on the TP affect precipitation in North China(Zhao et al. 2003). Zhou et al.(2009) also pointed out that, when the atmospheric heat source of the TP is strong, the low-lying layer in southern

China is dominated by anomalous southwesterly airflow, and northerly winds with low-level anomalies appear north of the Yangtze River, thus strengthening the low-level convergence of the Yangtze River basin. The convection in the East Asian and South Asian monsoon regions is relatively strong, with a large range of precipitation from Sichuan to the Yangtze River Delta.

It can be seen that, although domestic and foreign researchers agree that the heating effect of the

TP is one of the mechanisms affecting East Asian summer monsoon precipitation, the mechanism of how altitudinal heating affects summer monsoon precipitation remains debated. Xu et al. (2015) pointed out that the influence of solar radiation causes a "quick response" of the large-scale sensible heat of the TP and its relatively high-value dynamic movement. The front line of the midsummer rain belt just stops in the transition the mountain-plain transition zone in China, which suggests the TP may play a

key role in the land–ocean–atmosphere interaction of the summer monsoon process. In summer, the large-scale topography of the TP affects the movement of the summer rain belt. But what is the temperature change on the TP against the background of global warming? And how does the role played by the large-scale terrain affect the extent of warming? If the large-scale terrain is sensitive to global warming, will the influence of the TP and its uplift-related heating on atmospheric circulation be

magnified or reduced, and will it further affect the pattern of precipitation in East Asia?





## 2 Data and methods

### 2.1 Data

This study mainly uses the daily climate dataset (version 3.0) provided by the China Meteorological Science Data Network, including the daily maximum temperature, daily minimum temperature, and daily average temperature, at the cumulative times of 20:00–08:00, 08:00–20:00, and 20:00–20:00 hours. For precipitation, the data collection points include 824 national-level benchmarks and basic stations in the country, and the data age is 38 years from 1979 to 2016. Summer is June, July and August.

It is known that Global Precipitation Climatology Project (GPCP) data reflect the East Asian summer monsoon precipitation well (Adler et al. 2003), while the Modern-Era Retrospective Analysis for Research and Applications (MERRA) reanalysis is a dataset that represents the temperature variation of the TP successfully (Rienecker et al. 2011). Accordingly, monthly mean precipitation data from 1979 to 2015areobtainedfrom the GPCP dataset via https://www.esrl.noaa.gov/psd/data/gridded/data.gpcp.html, with a horizontal resolution of 2.5°×2.5°. Likewise, monthly mean MERRA data, with a horizontal resolution of 2/3° × 1/2° and covering the period 1979–2014, are obtained from https://www.disc.sci.gsfc.nasa.gov. Version 2 of the dataset (MERRA2), with a horizontal resolution of 0.625° × 0.5° and covering the period 1980–2016, is also used. These reanalysis data are used to analyze the atmospheric circulation associated with the warming of the TP and the summer precipitation in East Asia.

### 2.2 Methods

The ground-level climatological observation data are subject to quality control. The screening conditions include: (1) continuous data from 1979 to 2016; (2) extreme-value control; and (3) removal of stations with obvious migration (spherical displacement $d_{ij} \geq 20$ km or altitude shift $h_{ij} \geq 50$ m), calculated as follows:

$$d_{ij} = R \bullet a\cos(\sin\varphi_i \sin\varphi_j + \cos\varphi_i \cos\varphi_j \cos(\lambda_j - \lambda_i)) \tag{1}$$

$$h_{ij} = \left| h_j - h_i \right| \tag{2}$$

Where $\lambda$ is the latitude of the station, $\varphi$ is the longitude of the station, $h$ is the height, $R$ is the radius of the earth, and $i$ and $j$ are the year numbers of the data series.

In the eastern part of the TP, stations at altitudes above 2000 m are selected as representative



stations. After screening, there are a total of 61 stations in the eastern part of the TP and 693 stations

with precipitation data. The method of trend analysis employed in this study is calculated according to

the principle of least squares. The trend test method is the Mann–Kendal statistical test (Kundzewicz

and Robson, 2002), which is a nonparametric statistical test method widely used in linear trend testing.

   Regression analysis is a method used to find the statistical relationship between several variables,

including linear regression and nonlinear regression. Because the linear regression model is relatively

simple and rigorous in theory, this method is often used to analyze many meteorological elements in

meteorology (Piao et al. 2014; Zhang et al. 2019; Jo et al. 2019). Piao et al. (2014) used the linear

correlation to show the relationship between the Normalized Difference Vegetation Index (NDVI), a

proxy of vegetation productivity and temperature in northern ecosystems. They found that the strength

of the relationship between them (partial correlation coefficient) declined substantially between 1982

and 2011. The linear correlation is also used by Zhang et al (2019), and they identified that the widely

recognized inverse relationship (partial correlation coefficient) of central Eurasian spring snow cover

with the Indian summer monsoon rainfall has disappeared since 1990. Jo et al.(2019) found that a

positive sea surface temperature (SST)-precipitation relationship in the western tropical Pacific during

boreal spring, in which higher SSTs are associated with higher precipitation, episodically weakens from

the late 1990s to the early 2010s. In this paper, linear regression is also primary used to diagnosis the

relationship between East Asian Summer precipitation and the TP. So the results in this paper are based

on the assumption of the linearity of the complex systems. The correlation significance test is a bilateral

significance test considering the degrees of freedom.

**3. Temperature change over the TP and precipitation change over East Asia in summer**

**3.1 Temperature change over the TP and surrounding areas**

   Figure 1a shows the linear fit of summer time regional temperature during 1979–2016. A marked

increase is apparent in the past 40 years in the eastern part of the plateau, of which 56 stations show a

statistically significant trend at the 95% confidence level (solid red dots).There are 16 stations with a

trend exceeding0.5°C/decade, all distributed over the northeast and southwest sides of the TP, among

which Mangya Station (No.51886) even has a trend greater than 1.0°C/decade. From the interannual

variation of regional-average temperature in the eastern region of the TP (Figure 1b), the linear trend of

regional temperature is 0.39°C/decade(statistically significant at the 99% confidence level), which is far

greater than the rate of global warming. The Fifth Assessment Report of the Intergovernmental Panel on


Climate Change pointed out that the near 130-year (1880–2012) global-average surface temperature increased by 0.85°C.The2-m surface temperature from MERRA2 is also used as a comparison (blue line). During 1980–2016, the global-average surface temperature increased by 0.15°C/decade, which is also lower than the TP warming rate.

    Between 1979 and 2004, the interannual variability of temperature on the TP is relatively small

(except in 1981), basically remaining between 12.6°C and 14°C. After 2005, however, the overall temperature rise of the TP is significantly higher, at 13°C–14.8°C, and the oscillation amplitude is also large. Previous studies have shown that the warming rate of the TP is greater than in other regions of the world, including other plain regions of the same latitude in the Northern Hemisphere (Liu and Chen 2000; Wang et al. 2008; Liu et al. 2009; Wang et al. 2014).

The thermal effects of the TP have an important impact on the Asian monsoon and precipitation variability. The East Asian monsoon includes the South China Sea, the tropical Pacific monsoon region of the Northwest Pacific, and the subtropical monsoon region of the East Asian continent and Japan. It is located in the vicinity of the eastern part of the TP. These regions have monsoon troughs and high-rises with South Asian highs, and the ascending motion is more significant than in other regions.

The abnormal heating of the plateau plays an important role in the precipitation variability over East Asia. Wang et al. (2008) conducted experiments using the ECHAM4 global atmospheric model to show that plateau warming has a significant effect on the precipitation of the East Asian summer monsoon.

    In order to further analyze the temperature changes on the TP, the observational data and the MERRA reanalysis data of the TP and its surrounding areas (20°–40°N, 90°–110°E; dashed frame in

Figure 1a) are employed. Additionally, the 2-m air temperature data are further analyzed, and the results are shown in Figure 2. From the results of the two sets of data (Figures 2a and 2b), it is clear that the temperature of the TP and its surrounding areas decrease with increasing altitude, which is consistent with the variation of tropospheric temperature with height. However, it decreases at a rate of 0.43°C–0.45°C/100 meters, which is lower than the lapse rate (0.6°C/100 meters)in the tropospheric

atmosphere. This is very important to the interpolation of the temperature at ground stations on the TP. Previous work has often used 0.6°C/100 meters for interpolation when analyzing the plateau area (e.g. Zhou et al. 2009; Sun et al. 2013), making the surface temperature much lower, and possibly biasing the thermal effect of the TP.

    The temperature of the TP and its surrounding areas decreases with altitude by 0.43°–0.45°C/100

meters, which means that, for every 100 meters increase in height, the temperature over the TP is about





0.15°C higher than over other areas at the same level, which further indicates that the TP is a huge warm center in summer. However, against the background of global warming, how does the large-scale terrain of the TP affect the temperature change?

Figures 2c and 2d show the elevation-dependence of the warming rate in the observational and
reanalysis data over the TP and its surrounding areas. Apparently, the rate of temperature change of the observation stations at the altitude of 4000 meters is 0.4°C/decade/100 meters, while at stations at low altitudes of less than 500 meters it is about 0.2°C/decade, which is approximately0.2°C higher. In general, the MERRA reanalysis data also show an increasing elevation-dependent warming rate, although the warming rates with altitude differ from the observed one in magnitude. The linear fitting of
the two sets of data is significant, and all trends are statistically significant at the 95% confidence level. The results show that the TP is one of the most sensitive areas in terms of its response to global warming. The large-scale terrain of the TP has a magnifying effect on the warming rate of warm air, and the temperature increase in the high-altitude region is higher than that in the low-altitude region. This further confirms that the TP is possibly one of the most sensitive regions in terms of its response to
global warming (Liu and Chen 2000; Liu et al. 2009; Wang et al. 2014, Wu et al. 2017). The magnifying effect of the large-scale terrain under global warming might be related to the longwave radiation on the plateau and the albedo of clouds and snow, which needs further exploration (Liu et al. 2009; Wang et al. 2014)

The observational data and MERRA reanalysis data of the TP and its surrounding areas (20°–40°N,
90°–110°E) are divided into three levels according to altitude to analyze the probability density function of the temperature-change trend (°C/decade) in the region, as showed in Figure 3. These levels are: 0–2000 meters, 2000–4000 meters, and 4000–6000 meters. However, owing to the small number of stations at an altitude of 4000–6000 meters, the observational data are only divided into two levels. Comparing the two sets of data, the probability density function distribution of the temperature-change
rate is different. As shown in Figure 3a, the normal distribution curve of the 2000–4000meters temperature rate is significantly more concentrated and shifted to the right than that of 0–2000meters. The average temperature-change rate increases from 0.26°C/decade to 0.38°C/decade, and the variance reduces from 0.05 to 0.04. It can be seen that, with the increase of altitude, the temperature-change rate of the TP and its surrounding areas increases significantly. There is a small difference between the
MERRA reanalysis data analysis results and the observational data. The normal distribution curve of the temperature-change rate of 2000–4000 meters is non-significantly different from that of 0–2000 meters,



with only a slight shift to the right. The average temperature-change rate increases from 0.02°C/decade to 0.07°C/decade, and the variance does not change. However, the temperature curve of 4000–6000 meters shifts to the right obviously and becomes "short", which indicates the average

temperature-change rate increases and the variance increases, and the temperature-change rate of the high-altitude grid points is more discrete.

The results in Figure 3 verify the conclusions drawn from Figure 2: The TP region is one of the most sensitive in terms of its response to global warming; the large-scale terrain of the TP has a magnifying effect on the warming rate of warm air; and the warming is more prominent at higher than

at lower elevations.

### 3.2 Characteristics of summer precipitation in East Asia

Figure 4 shows the spatial distribution of precipitation linear fitting in China (a) and East Asia (b) during the summers of 1979–2016. In this period, the summer precipitation in China is characterized by the so-called "southern flood–northern drought" spatial distribution (Rectangular area from north to

south, respectively present north and south eastern Asia region). The precipitation in the region south of the Yangtze River, such as South China and Jiangsu and Zhejiang, is generally above normal and more pronounced, especially in Jiangsu and Zhejiang provinces. Precipitation in the region north of the Yellow River and in Southwest China is generally weaker. It is worth pointing out that the precipitation change in the south is more significant than in the north. The precipitation over the North China Plain

and Southwest China is slightly different, with more precipitation in the former and less obvious precipitation in the latter. In this paper, the monthly mean GPCP data are also used to present the variation in summer precipitation in East Asia. In China, the southern flood–northern drought pattern is also seen with the MERRA data, and is extremely significant. The similarity between the site data and MERRA data further confirms this southern flood–northern drought precipitation distribution pattern of

the summer monsoon in East Asia, which is consistent with the results of previous analyses (Xu et al. 2002; Wu et al. 2007; Wang et al. 2008; Xu et al. 2014). We also know precipitation is relatively stronger in the northern part of the South China Sea, the East China Sea, and the western North Pacific region, while weaker in Mongolia and the Sea of Japan. There are many factors affecting the summer monsoon precipitation in East Asia, such as the sea surface temperature, the northwestern Pacific

subtropical high, typhoons, TP heating, and so on. The influence of the thermal effects of the TP region on East Asian summer monsoon precipitation has been widely researched(Ye et al. 1992; Ding 1992;



Wu et al. 1998, 2012; Lu et al. 2005; Liu et al. 2012; Xu et al. 2008; Boos et al. 2013).

**3.3 Regression between TP warming and summer precipitation in East Asia**

The spatial distribution of regression analysis(Figure 5) is obtained by using the regional-average
temperature curve in the eastern part of the TP (Figure 1b) and the summer precipitation in East Asia.
Comparing Figure 4 and Figure 5, the regression pattern (Figure 5) resembles the leading pattern of East
Asian rainfall variations (Figure 4), suggesting East Asian summer monsoon rainfall might be related to
TP warming. The effects of TP warming are seen both locally and remotely. It can be seen that the TP
temperature is positively correlated with precipitation in eastern China, the northern South China Sea,
and northwestern Pacific, while negative correlation is found in Inner Mongolia, Northeast China,
Southwest China, and the Japan Sea. The remote impacts of TP warming are seen from the wave train
in Figure5, which moves along the low-level monsoon westerly, as also reported in Wang et al. (2008).
It is also apparent that the southern flood–northern drought pattern is more southerly, especially the
negative centers of the Japan Sea and southern plateau. When the plateau temperature is anomalously
high, rainfall might be stronger over southern Asia, weaker in the north, and vice versa.

The large-scale terrain of the TP has a magnifying effect on the warming rate of warm air, and the
temperature increase in the high-altitude region is higher than that in the low-altitude region. In order to
further analyze the influence of warming amplification on East Asian summer rainfall, regression
analysis with East Asian summer monsoon precipitation is also used. The curve is obtained by
removing the trend of temperature change in the eastern part of the TP, and the result is presented in
Figure 5b.Comparedwith Figure 5a, the spatial distribution is very similar; plus, the degree of regression
is better in most parts of East Asia, but especially in the low-level wave-train region. The difference
between them (Figure 5b minus Figure 5a; figure not shown)is also leading a resemble pattern with Fig
5b, and the low-level wavetrain is clear. After warming amplification, the southern flood–northern
drought pattern is weak. This indicates that TP warming has a certain relationship with this pattern, and
amplification of the TP warming may weaken it.

However, regression analysis only serves as a preliminary reflection of the phenomenon; further
analysis of the circulation characteristics, such as the water vapor conditions, is needed, along with
numerical simulation results, for verification.



## 4. Influence of TP warming on East Asian summer monsoon precipitation

### 4.1 Circulation over the TP in summer

The thermal effects of the TP have a significant impact on atmospheric circulation. Ye et al.(1957) pointed out that, in summer, the diabatic heating in the TP region induces a strong and stable
anticyclone at upper levels and a cyclone at lower levels, and so for hydrostatic balance the mid-troposphere must become warmer. The vertical movement is basically ascending, following strong convective activity over the TP region. Flohn (1957) also found that a "thermal anticyclonic cell" in the middle of the troposphere over the TP from July to August, inferring a relationship with the elevation heating of the plateau. Research on the uplifting (cooling) of the TP and its impact on atmospheric
circulation has from then on been extensive. Wu and Liu (2000) studied the thermal adaptation characteristics of atmospheric motion. The main points of thermal adaptation include: (1) the heat source in the atmosphere excites low-level cyclonic circulation and upper-level anticyclonic circulation; (2) the heating of the near-surface layer causes the isentropic surface to intersect the surface at the edge of the heating zone. According to the principle of potential vorticity conservation, a large amount of
negative vorticity is created for the gas column by friction, and the edge of the heating zone can be symmetrically unstable, with a deep anticyclone generated above the heat source zone.

It can be seen from Figure 6 that the 200-hPa wind field in the TP region presents a strong, zonally asymmetric, anticyclonic circulation, while the 700-hPa height field is controlled by a strong cyclonic circulation. In this configuration, there is a strong suction effect over the plateau, and the vertical
direction presents strong ascending motion, which is consistent with previous conclusions. From the asymmetric latitudinal temperature field, the high temperature center (356K) of the 200-hPa height field is consistent with the center of the flow field, both located above the TP and decreasing to the periphery, and the same as at 700hPa.In the summer, the TP is a strong and stable convective warming center from the low level (700hPa) to the upper level (200hPa). With the heat source center, the lower layer is a
strong cyclonic circulation, and the upper level is a strong anticyclonic circulation, which promotes upward movement. In fact, the strong uplift makes surrounding flows converge into the TP area and then propagate eastward, affecting the downstream. The analysis in Figure 2 also highlights that the plateau's large-scale terrain has an amplification effect on the warming around the surrounding area, and the warming rate in the high-altitude region is higher than that at low altitude.

But is an amplification effect of TP warming also found against the background of global warming?



Why does the amplification effect of TP warming weaken the southern flood–northern drought pattern of East Asian summer rainfall?

**4.2 Influence of TP warming on the circulation over East Asia**

The left-hand column (a, c, e) of Figure 7 shows the wind field at the upper level (200hPa), and the right-hand column (b, d, f) shows the same but at the lower level (700hPa). From the linear fitting of the wind field at 200hPa (Figure 7a), a strong anticyclonic circulation is apparent in the upper part of the northern TP, while the southwestern part of the plateau is characterized by cyclonic circulation. Corresponding to the low-level (700hPa) geopotential height field (Figure 7b), there is also the same feature: a strong anticyclonic circulation over Mongolia, covering the entire northern part of East Asia, and a cyclonic wind field over the southern part of East Asia. This means that, after 1979, the anticyclone circulation is gradually enhancing from the lower layer to the upper layer of Mongolia (with the cyclonic circulation weakening), indicating that the subsidence movement in the area is strengthened (ascending movement is weakened), which is not conducive to rainfall in the region. Meanwhile, the cyclonic circulation in the lower and upper levels of southern Asia indicates that the ascending motion is enhanced. In the presence of moisture feedback, the TP-warming-induced ascent to the east of the TP will induce more precipitation over East Asia. The distribution pattern is almost identical to the southern flood–northern drought pattern of East Asian summer rainfall, indicating that TP warming may be an important reason behind this pattern, warranting further discussion.

To analyze the influence of TP warming on atmospheric circulation, regression analysis between the TP temperature and wind fields is calculated separately, and then the two correlation coefficients are combined into a vector. Streamlines are made, and the results are presented in Figures 7c for 200 hPa and7d for 700 hPa. The correlation circulation field is transformed into a cyclonic circulation from north to south of the TP, and the high and low layers are consistent, despite the northern anticyclonic circulation and the southern cyclonic circulation weakening at 700hPa. It is clear, whether at 200hPa or700hPa, the circulation pattern is rather similar to the spatial distribution of wind change itself (Figures7a and 7b). The regression distribution (Figures 7e and 7f) after the summer temperature trend has been removed in the eastern part of the TP also bears this resemblance. TP warming is an important reason behind wind change at the low and upper levels.

To further explore the impact of TP warming amplification on East Asian summer monsoon precipitation, the differences between the regression of detrended temperature (Figures 7e and 7f) and





original temperature (Figures7c and d) are presented (Figure8). The influence of TP warming amplification is seen from two distinct Rossby wave trains—one in the extratropics that moves along the upper-level westerly jetstream (Figure8a), and one in the tropics that moves along the low-level monsoon westerly (Wang et al. 2008). Over the western North Pacific, a near barotropic anticyclone is found to the southwest of Japan, which is a part of the wavetrain induced by the TP. Accompanied by northerly wind from the low to the upper level, the southwest of Japan is mainly controlled by downdrafts, and moisture barely converges, meaning the southwest of Japan is anomalously dry in summer. The low-level cyclonic circulation over the northern South China Sea is a part of a wavetrain along the South Asian monsoon westerly that is easily excited in the southwest of the TP, increasing water vapor transport towards East Asian rainfall.TP warming induces the circulation structure, but amplification of TP warming weakens the anticyclonic circulation over Mongolia and the cyclonic circulation over Southeast Asia, and the southern flood–northern drought pattern is weak.

To further explore the difference of the water vapor condition between TP warming and warming amplification in terms of the effect on East Asian summer monsoon precipitation, regression analysis between TP temperature and water vapor flux is calculated separately, and then combined into a vector.

From the perspective of water vapor flux, when the plateau is warmer, whether at 500hPa or 700hPa, there is a clearer anomalous anticyclonic circulation over Mongolia. The water vapor flux is divergent at 500hPa and 700hPa, and northern Asia is dominated by downdrafts, which is not conducive to precipitation in the area. In Southeast Asia, on the contrary, over the southern part of China, the northern part of the South China Sea, and the northwestern Pacific, the water vapor flux is convergent at 500hPa, but weakly convergent at 700hPa.

After the TP warming trend has been removed, the influence of TP warming amplification is clear (Figure9). There are obvious water vapor wave trains that move along the low-level monsoon westerly. The low-level cyclonic circulation over the northern South China Sea is a part of a wavetrain along the South Asian monsoon westerly that is easily excited in the southwest of the TP, increasing water vapor transport towards East Asian rainfall. However, accompanied by divergent water vapor from low to upper levels, over the high-latitude TP and east of it, control is mainly exerted by downdrafts, and moisture barely converges, meaning the southwest of Japan is anomalously dry in summer. This further verifies the previous conclusions.

**5. Conclusions and discussion**

(1) The meteorological observation data show that, between 1979 and 2016, the temperature of the TP





increases at a relatively fast rate (0.39°C/decade), which is far greater than the rate of global warming. There are 16 sites with a trend of 0.5°C/decade, distributed on the northeast and southwest sides of the TP.

(2) The surface temperature of the TP and its surrounding areas decreases by 0.43°C–0.45°C/100meters, which is lower than the 0.6°C/100meters in the troposphere, and the temperature-warming rate varies with elevation. The magnitude of the temperature trend increases significantly with elevation, and warming amplification over the TP is an important and obvious feature of the temperature change. This suggests the TP is highly sensitive to global warming, and warming amplification might serve as an

early warning signal of global warming and climate change.

(3) TP warming is consistently the important contributor to the variation of East Asian precipitation in summer from 1979 to 2016, but the relationship between them weakens after the amplification of warming. The southern flood–northern drought pattern is weak compared with when the TP warming trend has been removed. In the past 40 years, the anticyclonic circulation in the upper and lower layers

over Mongolia in East Asia has strengthened obviously, while the southeastern part of China, the northern part of the South China Sea, and the northwestern Pacific have witnessed obvious cyclonic circulation anomalies. Such anomalous circulation is related to the TP warming. Under such a structure, the formation (at 700hPa) of divergence over Mongolia and the sinking airflow at the upper level (200hPa) are not conducive to precipitation in the area. The uplifting movement in the region is

strengthened in the Southeast Asia. This is also conducive to the entrapment of water vapor over the Bay of Bengal, South China Sea, and Northwest Pacific, providing dynamic conditions and water vapor conditions for precipitation.TP warming induces this circulation structure, but the amplification effect of the TP warming weakens its anomaly. The atmospheric response to TP warming might be related to Rossby waves. It is also found that two distinct Rossby wave trains and the isentropic uplift to the east

of the TP deform the western Pacific subtropical high and enhance moisture convergence towards southern East Asia, but less so in the northern TP.

It is important to note that results in this paper are based on the assumption of the linearity relationship between East Asian Summer precipitation and the TP temperature. This is a simplification of the complex climate-land-atmosphere systems. Now we are also trying to look into non-linear

methods to disclose further causality within the complex climate system, and we'll show it when the results come out. At the same time, only site-level data and reanalysis data are used for analysis in this study. To achieve a deeper understanding of the physical mechanisms and form a sound theoretical



basis, it remains necessary to also use numerical model results for further verification.

**Author contribution:**

Jianjun Xu and Liguang Wu designed the experiments and Mei Liang carried them out. Xiangde Xu put forward some useful modify opinions. Mei Liang prepared the manuscript with contributions from all co-authors."

**Acknowledgements:**

This research was jointly supported by the National Key R&D Program of China through grants 2018YFC1505705 and 2018YFC1505706, the Strategic Priority Research Program of Chinese Academy of Sciences through grant XDA20060503 and the State Key Laboratory of Natural Disasters, Fund key projects 41730961. The authors declare no competing financial interests. We are grateful for the daily temperature data and precipitation data provided by the China Meteorological Science Data

Sharing Network. We would also like to thank NOAA and NASA for providing the GPCP and MERRA reanalysis data.



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



540                                                    **Figure Captions**

Figure 1. (a) Temperature changes at stations above 2000 meters above sea level in the eastern TP
during the summers of 1979–2016 (units: °C/decade; solid points indicate statistical significance above
the 95% confidence level; the dashed frame is the study area). (b) Interannual temperature change
average over the TP (red)and globally(blue).Units: °C; **, statistically significant at the 95% confidence
level.

Figure 2.The (a, b) temperature (units:°C) and (c, d) temperature-change rate (units: °C/decade) over the
TP and its surrounding areas as a function of altitude: (a, c) observations; (b, d) MERRA. The red line is
the linear fitting curve; **, statistically significant above the 99% confidence level.

Figure 3. Distribution of probability density function of temperature-change rate (a, observational data;
b, MERRA data; units: °C/decade) in the TP and its surrounding areas. The black lines, red dotted lines
and blue dotted line indicate the altitude ranges of 0–2000 m, 2000–4000 m and 4000–6000 m,
respectively.

Figure 4. Spatial distribution of precipitation trends in (a) China (based on observations) and (b) East
Asia (based on GPCP data; bold contour delineates the altitude of 3000m) in the summers of
1979–2016 (units: mm/decade; cross symbols indicate statistical significance above the 95% confidence
level; Rectangular area from north to south, respectively present north and south east Asia region, the
same below).

Figure 5. Regression between (a) temperature in the eastern of the TP and summer precipitation from
1979 to 2016 (cross symbols indicate statistical significance above the 95% confidence level); (b) the
same as Fig. 5a but the temperature series trend have been removed in the eastern part of the TP.

Figure 6. Climatological distribution of the zonally asymmetric wind field (streamlines) and potential
temperature (units: K; red contours) over the TP from 1979 to 2014: (a, 200hPa; b, 700hPa).

Figure 7. The linear fitting of the wind field over East Asia during1979–2014 at (a) 200hPa and (b)
700hPa; Regression between summer temperature in the eastern TP and the wind field at (c) 200hPa
and(d) 700hPa; The same as Fig. 7c and 7dexcept that the summer temperature linear trend has been
removed in the eastern region at (e) 200hPa and(f) 700hPa.Shading indicates statistical significance
above the 95% confidence level.

Figure 8. Difference between the regression of the detrended temperature series with wind (vectors) and
vorticity(shaded)and the original temperature series with wind and vorticity at (a) 200hPa and(b)



700hPa.

Figure 9. Regression of TP warming with water vapor flux in East Asia at (a) 200hPa and (b) 700hPa, while(c, d) the same as Fig.9(a, b)respectively, but the trend of temperature series have been removed. (e, f) is the difference between (c, d) and (a, b) respectively. Shading indicates statistical significance above the 95% confidence level.

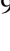



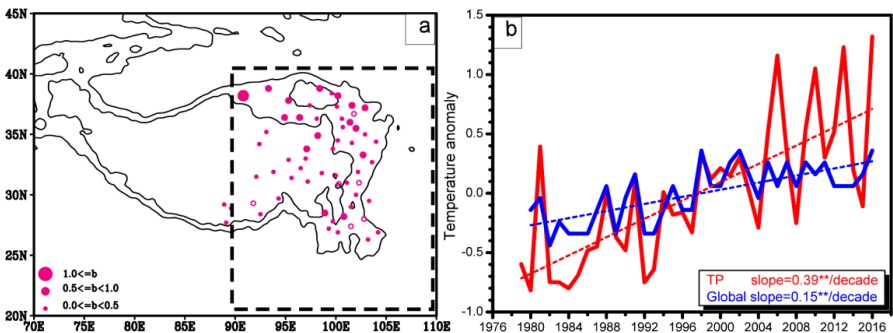

Figure1.(a) Temperature changes at stations above 2000 meters above sea level in the eastern TP during

the summers of 1979–2016 (units: °C/decade; solid points indicate statistical significance above the 95%

confidence level; the dashed frame is the study area). (b) Interannual temperature change average over

the TP (red)and globally (blue).Units: °C; **, statistically significant at the 95% confidence level.

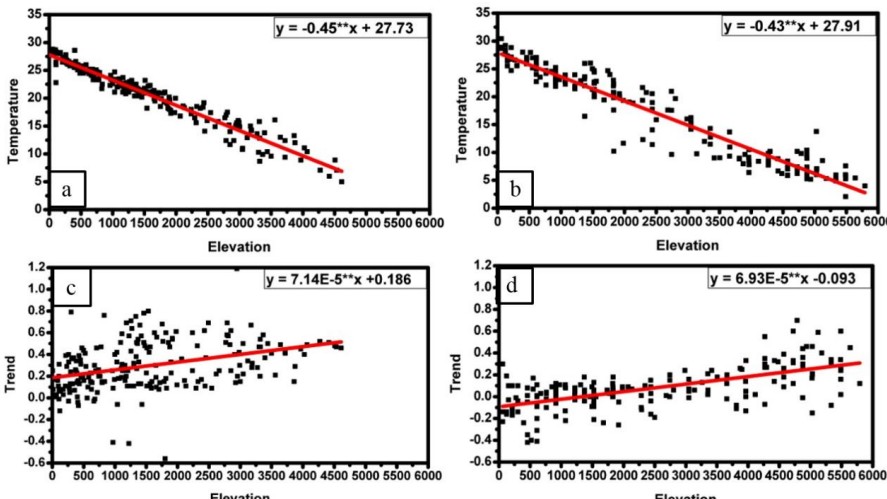

Figure 2. The (a, b) temperature (units:°C) and (c, d) temperature change rate (units: °C/decade) over

the TP and its surrounding areas as a function of altitude: (a, c) observations;(b, d) MERRA. The red

line is the linear fitting curve; **, statistically significant above the 99% confidence level.



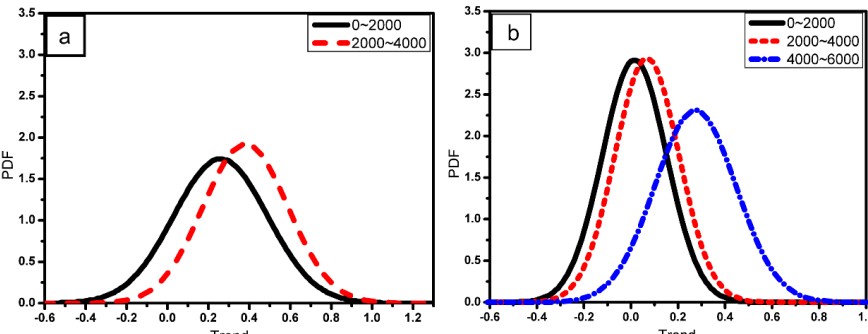

Figure 3. Distribution of probability density function of temperature-change rate (a, observational data; b, MERRA data; units: °C/decade) in the TP and its surrounding areas. The black lines, red dotted lines and blue dotted line indicate the altitude ranges of 0–2000 m, 2000–4000 m and 4000–6000 m, respectively.

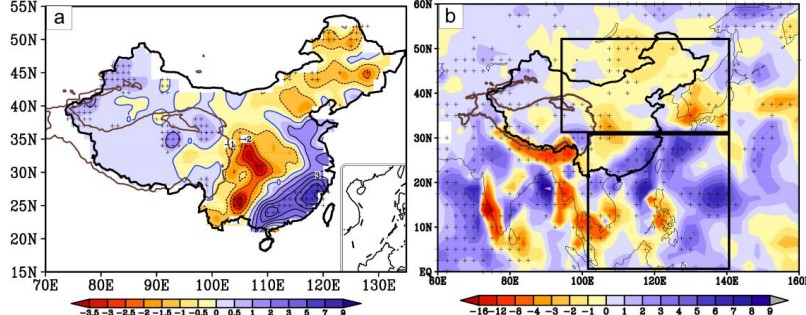


Figure 4. Spatial distribution of precipitation trends in (a) China (based on observations) and (b) East Asia (based on GPCP data; bold contour delineates the altitude of 3000m) in the summers of 1979–2016 (units: mm/decade; cross symbols indicate statistical significance above the 95% confidence level; Rectangular area from north to south, respectively present north and south east Asia region, the

same below).



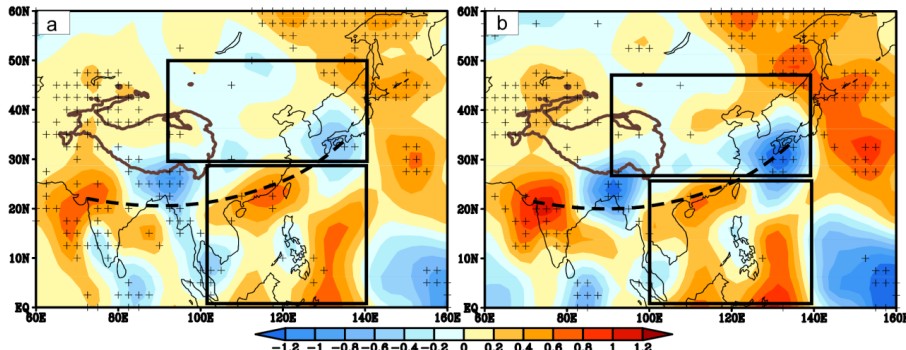

Figure 5. Regression between (a) temperature in the eastern of the TP and summer precipitation from 1979 to 2016 (cross symbols indicate statistical significance above the 95% confidence level);(b) the same as Fig.5a but the temperature series trend have been removed in the eastern part of the TP.

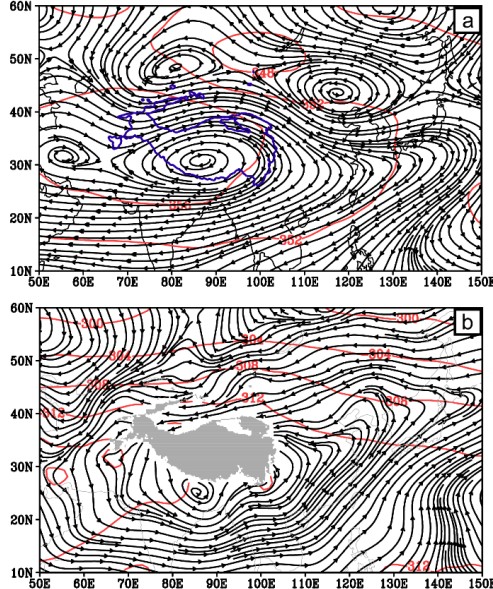

Figure 6. Climatologicaldistribution of the zonally asymmetric wind field (streamlines) and potential temperature (units: K; red contours) over the TP from 1979 to 2014 (a, 200hPa; b, 700hPa).



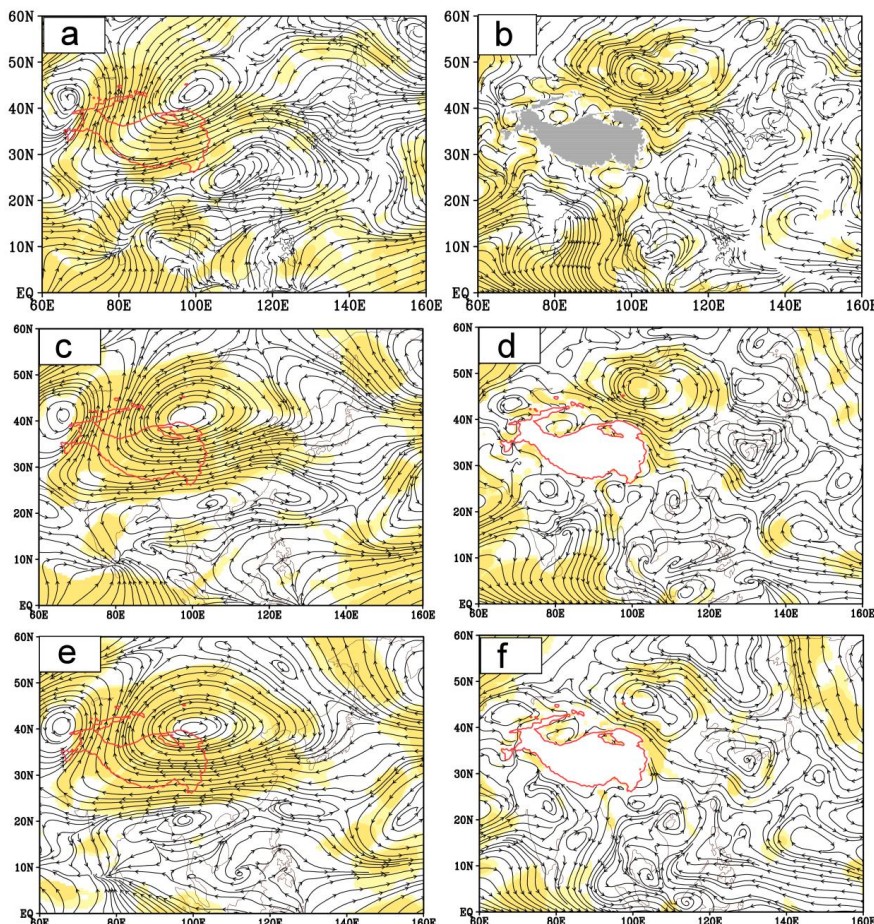

Figure 7. The linear fitting of the wind field over East Asia during1979–2014 at (a) 200hPa and(b) 700hPa;Regression between summer temperature in the eastern TP and the wind field at (c) 200hPa and(d) 700hPa; The same as Fig. 7c and 7dexcept that the summer temperature linear trend has been removed in the eastern region at (e) 200hPa and(f) 700hPa.Shading indicates statistical significance above the 95% confidence level.





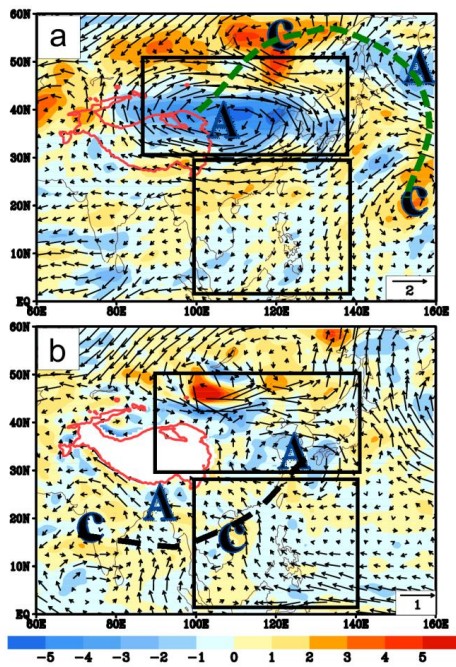

Figure 8. Difference between the regression of detrended temperature series and original temperature series with wind (vectors) and vorticity(shaded)at (a) 200hPa and (b) 700hPa.



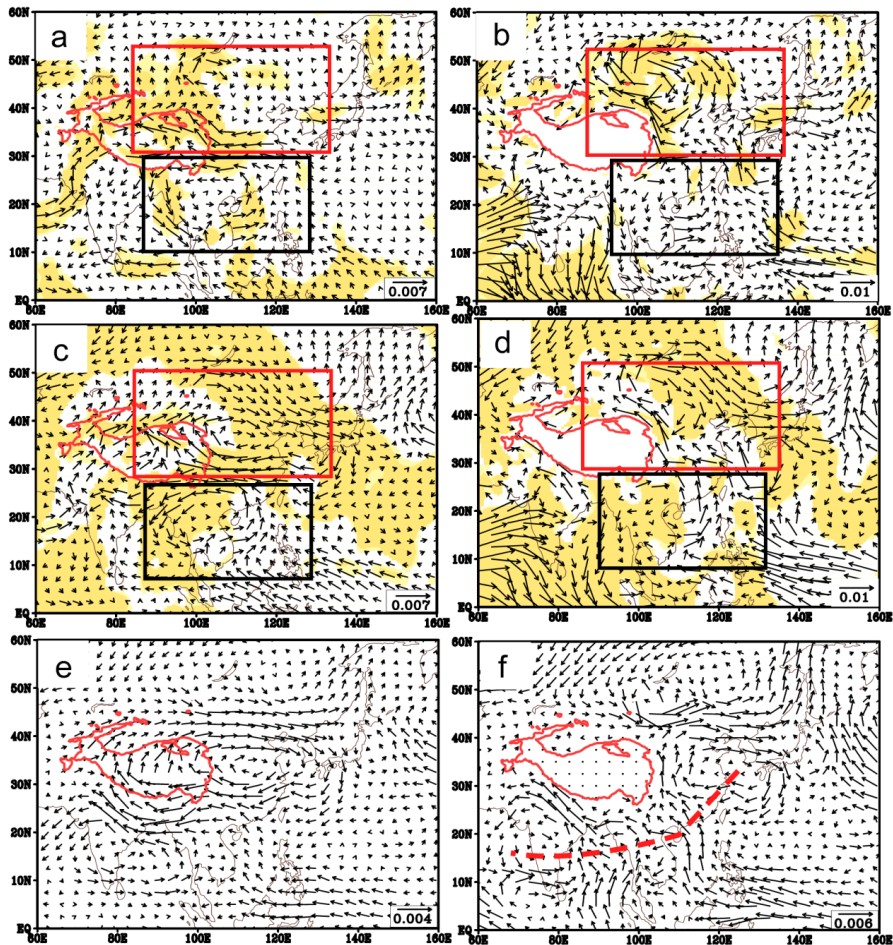

Figure 9. Regression of TP warming with water vapor flux in East Asia at (a) 200hPa and (b) 700hPa, while(c, d) the same as Fig.9 (a, b)respectively, but the trend of temperature series have been removed. (e, f) is the difference between (c, d) and (a, b) respectively. Shading indicates statistical significance above the 95% confidence level.