# Peer review of "Weakening anomalies of East Asian Summer Precipitation Influenced by the Tibetan Plateau Warming Amplification Mei Liang1,2, Jianjun Xu1,2, Liguang Wu3, Xiangde Xu4"

_Earth System Dynamics, 2019_

## Referee Comment (RC1) · Anonymous Referee #1 · 18 Jun 2019

Review of the "Weakening anomalies of East Asian Summer precipitation Influenced by the Tibetan Plateau Warming Amplification" by Mei Liang, Jianjun Xu, Xiangde Xu

This paper mainly addresses three key points: (1) the surface temperature-warming rate of TP area is greater than the rate of global warming; (2) the temperature change is related with elevation; (3) the TP warming is one of the factors which influence the East Asian summer monsoon. Linear regression analysis and Mann-Kendal trend test are adopted. Overall, the statistical methods used in this paper are not rigorous. For instance, the linear regression used in this paper does not provide the assessment of goodness of fit or validation. With this problem, all of the diagnosis which based on the regression are not acceptable. So, I would not recommend this paper unless the authors significantly improve their study.

Major Comments:

Lines 134-135: "In the eastern part of the TP, stations at altitudes above 2000 m are selected as representative stations." Why stations above 2000m are selected? Why only eastern part?

Line 139-146: Regression analysis and linear correlation are different, please double check the methodologies and cited literatures.

Line 157-170: Station based and region based trend analysis are done, together with some discussion on interannual variability. In fact, these results suggest that the study did not clarify which part is the trend, which part is the variability. Because the authors themselves also noticed and mentioned that "Between 1979 and 2004, the interannual variability of temperature on the TP is relatively small (except in 1981), basically remaining between 12.6°C and 14°C." Then before, the authors said "A marked increase is apparent in the past 40 years in the eastern part of the plateau, of which 56 stations show a statistically significant trend at the 95% confidence level (solid red dots). There are 16 stations with a trend exceeding 0.5°C/decade, all distributed over the northeast and southwest sides of the TP, among which Mangya Station (No.51886) even has a trend greater than 1.0°C/decade." These results made me wonder whether the trend analysis is reliable as probably there are abnormal

years that affect the trend analysis, especially, trend analysis is not robust when the data is short.

Line 186-190: "From the results of the two sets of data (Figures 2a and 2b), it is clear that the temperature of the TP and its surrounding areas decrease with increasing altitude, which is consistent with the variation of tropospheric temperature with height. However, it decreases at a rate of 0.43°C–0.45°C/100 meters, which is lower than the lapse rate (0.6°C/100 meters)in the tropospheric 190 atmosphere. " This statement is also based on the linear regression, assessments of goodness of fit shall be added. The same problem also exists in Line 199-201 and Line 222-223 and other places.

Line 207: "The large-scale terrain of the TP has a magnifying effect on the warming rate of warm air, …" I could not follow the authors conclusion. This might be speculation without evidence.

Line 214-218: Authors divide the altitude into three and two levels based on MERRA reanalysis data and observational data. However, there is no explanation on why they did this.

Line 220-224: "As shown in Figure 3a, the normal distribution curve of the 2000–4000 meters temperature rate is significantly more concentrated and shifted to the right than that of 0–2000 meters. The average temperature-change rate increases from 0.26°C/decade to 0.38°C/decade, and the variance reduces from 0.05 to 0.04." There is no statistical significant test, it is hard to believe the conclusion. In addition, authors shall make clear about how did they obtain the normal distribution of temperature.

Line 238-240: " In this period, the summer precipitation in China is characterized by the so-called "southern flood–northern drought" spatial distribution (Rectangular area from north to south, respectively present north and south eastern Asia region)." The interdecadal variations of summer precipitation over 1979-2016 in China has been investigated in many literatures. Authors should check the literatures and provide precise statement.

Line 247-248: "In China, the southern flood–northern drought pattern is also seen with the MERRA data, and is extremely significant. " what is the "extremely significant"? Any test to provide evidence.

Overall, the study relies on one single approach – linear regression/correlation. I don't think the linear method is sufficient to explain the complex system in Tibet and EASM. The study doesn't offer additional knowledge to the community, if the authors check the pool of literature regarding either EASM precipitation and climate variability of Tibet Plateau, they will find a lot of good studies that have been done years ago. Also, ESD, to our readers' understanding, emphasizes on the physical understanding of the system, the study does not offer deeper understanding.

---

## Referee Comment (RC2) · Anonymous Referee #2 · 22 Jun 2019

Major comments: Both the warming amplification over the Tibetan Plateau (TP) and the decadal change in eastern China feature by the so-called southern flood-norther drought have been well documented. In this work, the authors argue that the relationship between the TP temperature and summer precipitation in East Asia seems to be weakened after the enhanced warming amplitude over the TP. The data employed in this work, including station observed temperature and precipitation, GPCP precipitation field, together with MERRA2, are basically reliable. However, the method (liner regression) and overall procedure are questionable. Statistical relationship between the TP warming and summer precipitation in East Asia dese not necessary mean intrinsic connection between them. The explanation for the influence of TP warming upon

the summer rainfall change presented here cannot provide solid evidence. In fact, at least the following issues need be answered before one can accept the main conclusion drawn here. First, warming in which season is responsible for the circulation and rainfall pattern change in downstream regions? Second, during 1979-2016, for which period this connection is obvious? Third, if this connection is real, it appears in decadal time-scale or just linear trend? Fourth, what is the involved mechanism? In this work the authors claimed that the two Rossby wave trains related to the TP warming are responsible for the rainfall change in north part of East Asia and south part of East Asia, respectively. However, atmospheric wave pattern is stimulated by topography or diabatic heating, sometimes also generated from internal dynamics in atmosphere. Since the topography remains unchanged, TP warming induced heating anomaly or internal dynamics induces these two anomalous wave trains? Finally, global warming and/or interdecadal natural variability such as AO, PDO, and AMO are often used to explain the summer rainfall change in this area. How to exclude these factors and identified the regional contribution of the TP warming?

Specific comments:

1. The results shown in Fig.1 and 2 are annual mean or winter season? 2. Figure 4b. The two rectangular represent north and south parts of East Asia, respectively. However, salient regional difference in summer precipitation can be easily seen in these two domains. This basic feature has also been reported by many literatures. Therefore, it is not reasonable to divide the entire East Asia into only two regions. 3. Figure 5. What season for temperature in eastern TP? And it is also strange that the summer precipitation during 1979-2016 regressed on the temperature in eastern TP is almost same with that with liner trend removed. Did the author removes the trend in TP temperature and precipitation simultaneously? 4. Two Rossby wave trains shown in Fig.8a and b are used to explain the possible mechanism of TP warming effect. At least in the lower panel. i.e., the south branch in the lower troposphere, the wave pattern is hard to identify especially for the anticyclone just to the south of TP.

---

## Referee Comment (RC3) · Anonymous Referee #3 · 26 Jun 2019

The paper establishes 1) that the warming in the TP is elevation dependent and this warming is higher than the rate generally quoted for global warming, 2) The TP elevation-dependent warming is treated as a simple warming source for forcing precipitation variations in China and SE Asia, and 3) diagnose the changes of precipitation pattern in terms of circulation changes. It contributes to the understanding of the impact of TP in forcing the north-dry south-wet moisture pattern in China. The writing is understandable but clearly needs improvement. The approach uses linear regression for the trends and regression coefficients for examining relations. Vigor is lacking in some discussions, especially in the correlation fields. Two concerns on the paper: 1) to examine the impact of the TP warming, correlation pattern with and without the TP

trend is compared. It is not clear why the variance of due to the "total" TP time series is removed? 2) There are discussions on the relation of TP warming on global warming. The linkage has not been explicitly discussed. So jumping to the conclusion of the "most sensitive" feature is not warranted. There are inappropriate usage of the English language some of which I have tried to document below. I recommend publication after a rewrite to improve readability.

Minor comments: In Abstract Line 23 rewrite as: The present study documents the <effect of elevation-dependent temperature changes > on East Asian precipitation in summer over the Tibetan Plateau (TP). Line 25 Change <altitude> to <elevation>; Note: elevation refers to a place above sea level, altitude indicates an object above sea level. Line 26 change <troposphere> to <standard tropospheric lapse rate>. Delete <magnitude of the> and add <trend> after temperature. Line 28 change <the> to <an> Line 29 change <relations> to , change <amplify> to  Line 30 change <weak> to <weaken> , delete <compared with> Line 31 delete " amplification" Line 32 "rate of" Line 103 is "the coupling of the circulation and large scale terrain" sensitive to ... Line 110 indicate if this is GMT or Beijing time. Change at the "cumulative time" to " average over the period ... Line 140 change "including" to "it includes" simply references to linear regression. It has been widely used in meteorological statistical applications what is the purpose of quoting these examples, such as NDVI using linear regression? Delete these references unless you will refer to them later in the text. see Wilkes Line 170 compute the SD for these two periods to enable a quantitative comparison. Line 176 include "area" after "monsoon" Line 187 change altitude to elevation. Note: Lapse rate is the decrease in temperature with height. Change "tropospheric atmosphere" to "troposphere." Line 191 change altitude to elevation Line 204 change fitting to fits Line 206 how does this observation relates to global warming? State how did you show that this temperature change is due to global warming? Line 220 add "fit" after curve Line 221 change than to "compared to" Line 223 what is the unit of the SD? Again change altitude to elevation Lin 225 change to read "there are no significant differences between 0-2000m and 2000-4000m layer changes Line 233 again global

warming is invoked here, what is the supporting argument that this is due to global warming? Is there model simulation that show the magnification of the surface warming in the TP region? Even with supporting GHG simulation, one can only conclude that the data support the simulation? Line 237 add "change using" after "precipitation. Just state the means and SDs of the normal distribution fit. You can use a t test to check if there is a significant difference. Line 240 change "present" to "showing" Line 241,242 what is meant by "above normal and pronounce?" is it simply "higher?" Line 252 change "stronger" and "weaker" to "higher" and "lower" Line 258 change Regression to Relation, replace the first sentence with "Summer precipitation in East Asia has been regressed against the regional-average temperature (see Fig 1b) and the regression coefficients are presented in Figure 5. Line 275 by change to read "removing the linear trend of temperature," and delete "change." Why only the trend is removed and not the trend itself? Line 280 weak to weaker Line 309 "change is" to "has" Line 320 how do you do linear fitting of the wind field? Are these streamlines? line 360 are these vector fields significant? Should probably not include any points when the correlation is not significant. Line 383 the link between the TP warming and global warming is not clear. Also make changes in the figure captions to correspond to those in the text.
* * *

---

## Author Comment (AC1) · 14 Sep 2019

**Responses to Reviewer #3:**

*The paper establishes 1) that the warming in the TP is elevation dependent and this warming is higher than the rate generally quoted for global warming, 2) The TP*

*elevation-dependent warming is treated as a simple warming source for forcing precipitation variations in China and SE Asia, and 3) diagnose the changes of precipitation pattern in terms of circulation changes. It contributes to the understanding of the impact of TP in forcing the north-dry south-wet moisture pattern in China. The writing is understandable but clearly needs improvement. The approach uses linear regression for the trends and regression coefficients for examining relations. Vigor is lacking in some discussions, especially in the correlation fields. Two concerns on the paper: 1) to examine the impact of the TP warming, correlation pattern with and without the TP trend is compared. It is not clear why the variance of due to the "total" TP time series is removed? 2) There are discussions on the relation of TP warming on global warming. The linkage has not been explicitly discussed. So jumping to the conclusion of the "most sensitive" feature is not warranted. There are inappropriate usage of the English language some of which I have tried to document below. I recommend publication after a rewrite to improve readability.*

**Response:** Thanks again for the reviewer's valuable comments and suggestions that allowed us to improve the manuscript substantially. We have carefully considered these comments and suggestions, and revised the manuscript thoroughly.

(1) We think that the remaining part after the removal trend represents the interannual variability. We first detrend all quantities to allow focused analysis on correlations in interannual variability. In the remaining section, we believe that it contains long-term trend signals and interdecadal signals. To examine whether the impact of the TP warming is annual variability or trend, correlation pattern with and without the TP trend is compared. In the revised manuscript, decadal change is analyzed too.

(2) The relationship between TP warming and global warming is not clear yet.

Originally we just wanted to express TP warming is a special phenomenon under the global warming. In the first manuscript, our expression is unclear. We have reviewed the relevant literature and reorganized the sentences to express more correctly. The grammar has been thoroughly tidied up in the revision. In the correlation fields, we have added more detailed discussion. Thanks for your comments and suggestions, which effectively improved our manuscript. Please see below for our point-by-point response. The original comments are quoted in *Italic*.

**Minor comments:**

***Comment 1:*** *In Abstract Line 23 rewrite as: The present study documents the <effect of elevation-dependent temperature changes > on East Asian precipitation in summer over the Tibetan Plateau (TP).*

**Response:** Thank you for your advice. Revised as suggested.

***Comment 2****: Line 25 Change <altitude> to <elevation>; Note: elevation refers to a place above sea level, altitude indicates an object above sea level. Line 26 change <troposphere> to <standard tropospheric lapse rate>. Delete <magnitude of the> and add <trend> after temperature. Line 28 change <the> to <an>. Line 29 change <relations> to , change <amplify> to . Line 30 change <weak> to <weaken>, delete <compared with>. Line 31 delete "amplification". Line 32 "rate of" Line 103 is "the coupling of the circulation and large scale terrain" sensitive to: : :*

**Response:** Thank you for your advice. Revised as suggested.

***Comment 3****: Line 110 indicate if this is GMT or Beijing time. Change at the "cumulative time" to "average over the period".*

**Response:** Thank you for your advice. Revised as suggested.

***Comment 4****:* Line 140 change "including" to "it includes" simply references to linear regression. It has been widely used in meteorological statistical applications. what is the purpose of quoting these examples, such as NDVI using linear regression? Delete these references unless you will refer to them later in the text.

**Response:** Thank you for your advice. Revised as suggested, and we have delete these references.

***Comment 5****:* See Wilkes Line 170 compute the SD for these two periods to enable a quantitative comparison.

**Response:** Thank you for your advice. "To get quantitative comparison, Mann-Kendall non-parametric test and Theil-Sen estimator are both used, and SD for these two periods are also added. Both methods show that the trend for these two periods are tested at 99% level of significance, but the trend and SD of TP temperature are both larger than global warming. To check whether the trend of TP temperature is reliable or not, Mann-Kendal method is used to test whether there is a regime shift in TP temperature. Result showed that a regime shift is found in 1997, and it is tested statistically significant at the 95% confidence level. Then the time is broken into two period, 1980-1996 and 1997-2016. Both methods show that the trend for these three periods are tested at a certain level of significance." The above results and analysis have been supplemented in the paper. (Figure 2, P3, L194-204)

[Figure]

Figure 2. (a) Interannual change of summer temperature averaged by the TP (red) and global (blue). Trend significant test are from Mann-Kendall test (blue line) and the Theil-Sen estimator (green line). (Units: °C). (b) Mann-Kendall regime shift test of the temperature of the eastern part of the TP. Both trends are statistically significant at the 99% confidence level.

[Figure]

Figure S2. Trend analysis by Mann-Kendall test (blue line) and the Theil-Sen estimator (green line) for 1980-2016, 1980-1997 and 1998-2016 (unit: °C; red line) respectively. Statistically significant at the 95%, 94%, 99% confidence level respectively.

*Comment 6*: *Line 176 include "area" after "monsoon" Line 187 change altitude to elevation. Note: Lapse rate is the decrease in temperature with height. Change "tropospheric atmosphere" to "troposphere." Line 191 change altitude to elevation. Line 204 change fitting to fits.*

**Response:** Thank you for your advice. Revised as suggested.

*Comment 7*: *Line 206 how does this observation relates to global warming? State how did you show that this temperature change is due to global warming?*

**Response:** Thanks for your comments. We apologize for our over-interpreted. We have reviewed the relevant literature and reorganized the sentences as following: "Various reasons have been related to the elevation-dependent warming. Common explanation is associated with the snow-albedo feedback mechanism (Giorgi et al. 1997; Pepin and Lundquist 2008; Ceppi et al. 2010). Moreover, the cloud-radiation effects (Liu et al. 2009), water vapor and radiative fluxes (Rangwala 2013), are also responsible for it. " (P3, L255-258)

Reference:

Ceppi P., Scherrer S., Fischer A., Appenzeller C.: Revisiting Swiss temperature trends 1959–2008, Int J Climatol 32:203–213, https://doi.org/10.1002/joc.2260 , 2010

Giorgi F., Hurrel J., Marinucci M., Beniston M.: Elevation dependency of the surface climate change signal: a model study. J Clim 10:288–296. https://doi.org/10.1175/1520-0442 , 1997.

Liu X., Cheng Z., Yan L., and Yin Z.: Elevation dependency of recent and future minimum surface air temperature trends in the Tibetan Plateau and its surroundings. Global Planet Chang, 68:164–174, https://doi.org/10.1016/j.gloplacha.2009.03.017, 2009.

Pepin N., Lundquist J.: Temperature trends at high elevations: patterns across the globe. Geophys Res Lett 35: L14701. https://doi.org/10.1029/2008GL034026, 2008.

Rangwala I.: Amplified water vapor feedback at high altitudes during winter. Int J Climatol 33:897–903. https://doi.org/10.1002/joc.3477 , 2013.

***Comment 8****: Line 220 add "fit" after curve Line 221 change than to "compared to" Line 223 what is the unit of the SD? Again change altitude to elevation. Just state the means and SDs of the normal distribution fit. You can use a t test to check if there is a significant difference.*

**Response:** Thanks for your advice. The unit of the SD is °C/decade. We have added the statistical significant test, and the means and SDs of the normal distribution fit are also added in the upper right corner of the diagram.

"Kolmogorov-Smirnov test is applied to check the statistical significance of normal distribution. If the computed p-value is greater than the significance level alpha=0.05, one cannot reject the null hypothesis that the sample follows a normal distribution. "(P2, L175-177).

"The statistical significance of climate mean analysis is determined by the standard two-tailed Student's t test method. "(P2, L184-185).

"As shown in Figure 4a and 4b, the normal distribution curve fit of the 2000–6000 meters temperature rate is significantly more concentrated and shifted to the right compared to that of 0–2000meters. The average temperature-change rate increases from 0.26°C/decade to 0.39°C/decade (p<0.001). It can be seen that, with the increase of elevation, the temperature-change rate of the TP and its surrounding areas increases significantly. There is the same phenomenon between the MERRA reanalysis data and the observational data despite difference in value. There are also significant differences between 0-2000 and 2000-4000m layers changes. The average temperature-change rate increases from 0.02°C/decade to 0.07°C/decade (p<0.001). Between 2000-4000 and 4000-6000m, the average temperature-change rate increases from 0.07°C/decade to 0.28°C/decade (p<0.001). " The result is agreed with previous

conclusion. The results and analysis above have been supplemented in the paper. (Figure 4; P2, P3, L244-252)

[Figure]

Figure 4. Distribution of probability density function of temperature trend (a, b, observational data; c, d, e, MERRA data; units: °C/decade) in the TP and its surrounding areas. The histogram represents the samples size, and the curve is a normal distribution fit line. The means and SDs of the normal distribution fit are also added in the upper right corner of the diagram. (a), (b) indicates the altitude ranges of 0–2000 m, and 2000–6000 m, respectively. (c), (d), (e) indicates the altitude ranges of 0–2000 m, 2000–4000 m and 4000–6000 m, respectively. All graphs are statistically significant of normal distribution at the 95% confidence level by Kolmogorov-Smirnov test except (c). All mean tests are passed by student t test.

***Comment 9****: Line 225 change to read "there are no significant differences between 0-2000m and 2000-4000m layer changes.*

**Response:** Thank you for your advice. Revised as suggested.

***Comment 10****: Line 233 again global warming is invoked here, what is the supporting argument that this is due to global warming? Is there model simulation that show the magnification of the surface warming in the TP region? Even with supporting GHG simulation, one can only conclude that the data support the simulation?*

**Response:** Thanks for your comments. We apologize for our over-interpreted. We have reviewed the relevant literature and reorganized the sentences as following: "Various reasons have been related to the elevation-dependent warming. Common explanation is associated with the snow-albedo feedback mechanism (Giorgi et al. 1997; Pepin and Lundquist 2008; Ceppi et al. 2010). Moreover, the cloud-radiation effects (Liu et al. 2009), water vapor and radiative fluxes (Rangwala 2013), are also responsible for it. " (P3, L255-258)

Reference:

Ceppi P., Scherrer S., Fischer A., Appenzeller C.: Revisiting Swiss temperature trends 1959–2008, Int J Climatol 32:203–213, https://doi.org/10.1002/joc.2260 , 2010

Giorgi F., Hurrel J., Marinucci M., Beniston M.: Elevation dependency of the surface climate change signal: a model study. J Clim 10:288–296. https://doi.org/10.1175/1520-0442 , 1997.

Liu X., Cheng Z., Yan L., and Yin Z.: Elevation dependency of recent and future minimum surface air temperature trends in the Tibetan Plateau and its surroundings. Global Planet Chang, 68:164–174, https://doi.org/10.1016/j.gloplacha.2009.03.017, 2009.

Pepin N., Lundquist J.: Temperature trends at high elevations: patterns across the globe. Geophys Res Lett 35: L14701. https://doi.org/10.1029/2008GL034026, 2008.

Rangwala I.: Amplified water vapor feedback at high altitudes during winter. Int J Climatol 33:897–903. https://doi.org/10.1002/joc.3477 , 2013.

**Comment 11**: *Line 237 add "change using" after "precipitation". Line 240 change "present" to "showing" Line 241,242 what is meant by "above normal and pronounce?" is it simply "higher?" Line 252 change "stronger" and "weaker" to "higher" and "lower"*

**Response:** Thank you for your advice. Revised as suggested.

**Comment 12**: *Line 258 change Regression to Relation, replace the first sentence with "Summer precipitation in East Asia has been regressed against the regional-average temperature (see Fig 1b) and the regression coefficients are presented in Figure 5.*

**Response:** Thank you for your advice. Revised as suggested. (P3, L48-51)

**Comment 13:** Line 275 by change to read "removing the linear trend of temperature," and delete "change." Why only the trend is removed and not the trend itself? Line 280 weak to weaker Line 309 "change is" to "has".

**Response:** Thank you for your advice. Revised as suggested. I am very sorry that our statement is not clear. Here we removed is the trend itself.

**Comment 14:** *Line 320 how do you do linear fitting of the wind field? Are these streamlines? Line 360 are these vector fields significant? Should probably not include any points when the correlation is not significant.*

**Response:** Linear fitting of zonal wind and meridional wind is calculated separately, and then the two regression coefficients are combined into a vector. Previous manuscript are performed with streamlines instead of arrow. The revised manuscript, we indicated it with an arrow (Figure 6). If it is not including any points when the correlation is not significant, pictures may show certain discontinuities and may not identify the pattern. So, we plot all points, and shading the significant point.

***Comment 15:*** *Line 383 the link between the TP warming and global warming is not clear. Also make changes in the figure captions to correspond to those in the text.*

**Response:** Thanks for your comments. (1) The same problem has occurred several times, and we apologize for our over-interpreted. We have reviewed the relevant literature and reorganized the sentences as following: "Various reasons have been related to the elevation-dependent warming. Common explanation is associated with the snow-albedo feedback mechanism (Giorgi et al. 1997; Pepin and Lundquist 2008; Ceppi et al. 2010). Moreover, the cloud-radiation effects (Liu et al. 2009), water vapor and radiative fluxes (Rangwala 2013), are also responsible for it. " (P3, L255-258)

(2) We have checked several times and changed the figure captions to correspond to the title in the text.

Reference:

Ceppi P., Scherrer S., Fischer A., Appenzeller C.: Revisiting Swiss temperature trends 1959–2008, Int J Climatol 32:203–213, https://doi.org/10.1002/joc.2260 , 2010

Giorgi F., Hurrel J., Marinucci M., Beniston M.: Elevation dependency of the surface climate change signal: a model study. J Clim 10:288–296. https://doi.org/10.1175/1520-0442 , 1997.

Liu X., Cheng Z., Yan L., and Yin Z.: Elevation dependency of recent and future minimum surface air temperature trends in the Tibetan Plateau and its surroundings. Global Planet Chang, 68:164–174, https://doi.org/10.1016/j.gloplacha.2009.03.017, 2009.

Pepin N., Lundquist J.: Temperature trends at high elevations: patterns across the globe. Geophys Res Lett 35: L14701. https://doi.org/10.1029/2008GL034026, 2008.

Rangwala I.: Amplified water vapor feedback at high altitudes during winter. Int J Climatol 33:897–903. https://doi.org/10.1002/joc.3477 , 2013.

---

## Author Comment (AC2) · 14 Sep 2019

**Responses to Reviewer #2:**

We wish to thank the reviewer for the detailed comments and suggestions that helped us substantially improve the formula, phrasing, logic, and quality of our manuscript. We have revised the manuscript accordingly. Please see below our point-by-point response. The original comments are formatted in *Italics*.

**Major comments:** *Both the warming amplification over the Tibetan Plateau (TP) and the decadal change in eastern China feature by the so-called southern flood-norther drought have been well documented. In this work, the authors argue that the relationship between the TP temperature and summer precipitation in East Asia seems to be weakened after the enhanced warming amplitude over the TP. The data employed in this work, including station observed temperature and precipitation, GPCP precipitation field, together with MERRA2, are basically reliable. However, the method (liner regression) and overall procedure are questionable. Statistical relationship between the TP warming and summer precipitation in East Asia dese not necessary mean intrinsic connection between them. The explanation for the influence of TP warming upon the summer rainfall change presented here cannot provide solid evidence. In fact, at least the following issues need be answered before one can accept the main conclusion drawn here.*

***Comment 1:*** *First, warming in which season is responsible for the circulation and rainfall pattern change in downstream regions?*

**Response:** Thank you for your valuable comment. Note that, in the revised manuscript, MERRA2 reanalysis data is used to analyze the TP temperature and atmospheric circulation, so the study period was changed to 1980-2016. Figure S4 shows the standard partial regression coefficient of TP-EASM in 1980–2016 at spring (MAM), summer (JJA), fall (SON), winter (DJF) respectively. In Summer (Figure S4-a2), the spatial distribution of the partial regression coefficient like a "sandwich". Positive relationship is found in South China, Indochina Peninsula, South China Sea, and the Northwest Pacific Ocean. It's also obviously positively correlated North China and Northeast China too. Negative relationship is found along a

southwest-northeast oriented rain belt extending from southern TP across the mid-lower reach of the Yangtze River Valley to the Korean peninsula and Japan, which is primarily along the East Asia subtropical front (called Meiyu in Chinese). Comparing these four seasons (Figure S4), such a complete spatial distribution of the partial regression coefficient in East Asia, we found that only summer season. At the same time, the central value of the partial regression coefficient in summer can reach 0.5 and -0.6, and it is larger than that in other seasons. Consequently, we initially think that mainly summer season is responsible for the circulation and rainfall pattern change in downstream regions. The results and analysis above have been supplemented in the paper. (Figure S4; P3, L316-323)

[Figure]

[Figure]

Figure S4. The standard partial regression coefficient of precipitation and TP temperature in East Asia in 1980–2016 (precipitation based on GPCP data; brown contour delineates the altitude of 2000m). (a) represents all variables are not detrended; (b), precipitation and TP temperature have been detrended; (c) is the difference between (a) and (b); 1, 2, 3, 4 represent spring (MAM), summer (JJA), fall (SON), winter (DJF) respectively; black spots indicate statistical significance above the 90% confidence level.

**Comment 2:** *Second, during 1979-2016, for which period this connection is obvious?*

**Response:** Thanks for your valuable suggestion. Note that, in the revised manuscript,

MERRA2 reanalysis data is used to analyze the TP temperature and atmospheric circulation, so the study period was changed to 1980-2016. In order to check, during 1980-2016, which period this connection is obvious. At first, Mann-Kendal method is used to test whether there is a regime shift in TP temperature, and the result is showed in Figure 2. A regime shift is found in 1997, and it is statistically significant at the 95% confidence level. Then the entire research time is broken into two period, 1980-1996 and 1997-2016. The partial regression coefficient of precipitation and TP temperature in 1980-1996, 1997-2016 and 1980-2016 respectively (Figure 6). It shows that over the past 20 years (1997–2016) the relationship pattern of precipitation and TP temperature is change comparing with the period 1980-1996. In 1997–2016, negative relationship is found along a southwest-northeast oriented rain belt extending from the mid-lower reach of the Yangtze River Valley across South Korea to eastern Japan, which is primarily along the EA subtropical front (called Meiyu in Chinese, Changma in Korean, and Baiu in Japanese). Positive relationship is found in South China, Indochina Peninsula, South China Sea, and the Northwest Pacific Ocean. During 1980-2016, for the second period 1997-2016 this connection between TP warming and rainfall pattern change in downstream regions is obvious. The above results and analysis have been supplemented in the revised manuscript. (Figure 2 and Figure 6; P2, L293-310)

[Figure]

Figure 2. (a) Interannual change of summer temperature averaged by the TP (red) and global (blue). Trend significant test are from Mann-Kendall test (blue line) and the Theil-Sen estimator (green line). (Units: °C). (b) Mann-Kendall regime shift test of the temperature of the eastern part of the TP. Both trends are statistically significant at the 99% confidence level.

[Figure]

Figure 6. The standard partial regression coefficient of precipitation and TP temperature in East Asia. (a) represents the first period 1980-2016; (b) represents the second period 1997-2016; (c) is the whole period 1980-1996; (d) the same as Figure 6c but after detrending all data. black spots indicate statistical significance above the 90% confidence level.

***Comment 3***: *Third, if this connection is real, it appears in decadal time-scale or just linear trend?*

**Response:** Thanks for your suggestion. To obtain further insight into the connection of the temperature-monsoon relationship, we perform the partial regression coefficient of precipitation and TP temperature in different condition, as shown in Figure 6. It's believed that after TP warming amplification, the magnitude of the TP-EASM correlation enhance (strengthened positive correlations and strengthened negative correlations; Figure 6c). So, TP-EASM relationship is negative during the period of 1980–1996 but becomes obvious positive for the period of 1997–2016 in the South China Sea and the Northwest Pacific Ocean (Figure 6a and Figure 6b). At the same time, negative correlation along the East Asia subtropical front (Meiyu) become stronger in period of 1997–2016. In summary, during 1980-2016, this connection between TP-EASM is obvious after late 20 years, and the connection of it underwent obvious interdecadal variability and linear trend. The above results and analysis have been supplemented in the revised manuscript. (P3, L306-314)

[Figure]

Figure 6. The standard partial regression coefficient of precipitation and TP temperature in East Asia. (a) represents the first period 1980-2016; (b) represents the second period 1997-2016; (c) is the whole period 1980-1996; (d) the same as Figure 6c but after detrending all data. black spots indicate statistical significance above the 90% confidence level.

**Comment 4**, *what is the involved mechanism? In this work the authors claimed that the two Rossby wave trains related to the TP warming are responsible for the rainfall change in north part of East Asia and south part of East Asia, respectively. However, atmospheric wave pattern is stimulated by topography or diabatic heating, sometimes also generated from internal dynamics in atmosphere. Since the topography remains unchanged, TP warming induced heating anomaly or internal dynamics induces these two anomalous wave trains?*

**Response:** Thank you for your suggestions. In our study, we just performed the

statistical method (multiple regression method) to analysis the phenomenon instead of numerical model. But in previous research, some works have done, and we have quoted the results of them. Wang et al. (2008) performed numerical experiments with a comprehensive AGCM (ECHAM4) to see the mechanisms by which the atmosphere responds to the Tibetan Plateau warming. The mechanism through which the atmosphere responds to the warming TP is illustrated by the schematic diagram shown in Figure S5. Two Rossby wave trains are excited due to the TP warming. One has a barotropic structure and propagates downstream along the upper-level westerly jet stream to enhance the anticyclonic circulation to east of Japan. Another wave train developing along the low-level southwesterly monsoon propagates into the South China Sea and enhances the low-level anticyclonic ridge there.

Uncertainties remain in understanding the linkage between TP temperature and EASM precipitation based on statistical analysis. To achieve a deeper understanding of the physical mechanisms, it remains necessary to use numerical model results for further verification. We will check it by using CMIP6 model data and different numerical model in later work. Thanks for your valuable comment again.

[Figure]

Figure S5. Schematic diagram showing the mechanisms by which the atmosphere responds to the Tibetan Plateau warming, in particular the remote impact of TP warming on East Asian summer monsoon rainfall through two Rossby wavetrains. The letters A and C denote anticyclonic and cyclonic circulation centers, respectively (Wang et al., 2008).

Reference:

Wang B., Qing B., Brian Hoskins et al.: Tibetan Plateau warming and precipitation changes in East Asia, Geo. Res. Let. 35L14702, https://doi:10.1029/2008GL034330, 2008.

***Comment 5***, *global warming and/or interdecadal natural variability such as AO, PDO, and AMO are often used to explain the summer rainfall change in this area. How to exclude these factors and identified the regional contribution of the TP warming?*

**Response:** Thank you for your suggestions. In addition to TP temperature, one of the key factor affect East Asian summer monsoon (EASM) is the El Niño-Southern Oscillation (ENSO; Wu et al. 2003); on decadal scales, ENSO's influence on the monsoon can also be heavily modulated by the Pacific Decadal Oscillation (PDO; Feng et al. 2014; Song and Zhou 2015). Yet, the variability of the tropical Pacific can explain only part of the rich structure of the monsoon. Recently, remote teleconnections on East Asia from the North Atlantic region have garnered considerable attention. For example, the Atlantic Multidecadal Oscillation (AMO) has been linked to multidecadal changes in atmospheric circulation and precipitation over China (Lu et al. 2006; Lin et al. 2016; Wu et al. 2016a, b). The western North Pacific subtropical high (WNPSH) is an also crucial component of the East Asian summer monsoon (EASM) system and significantly influences the precipitation in East Asia (Lu, 2001; Lee et al., 2013). On the whole, global warming and interdecadal natural variability such as ENSO, NPSH, PDO, and AMO also influence the summer rainfall change in this area. Here multiple regression method is applied to exclude these factors. As a predictive analysis, the multiple linear regression model is used to explain the relationship between one continuous dependent variable and two or more independent variables. We just analysis the standardized partial regression coefficients between TP temperature and EASM precipitation, and evaluating the relative contribution to the EASM precipitation.

Comparing with single linear regression, it is some difference after considering other factors. In Figure 6, the results indicate that the strength (correlation) of the

linkage between TP-EASM (strengthened positive correlations in the south of East Asia and strengthened negative correlations in East Asia subtropical front) has increased with the TP warming amplification. Consequently, in the later period of 1997-2016, the spatial distribution of the correlation of TP-EASM relationship is change. It is from negative to obvious positive during the period of 1980–1996 to 1997–2016 in the South China Sea and the Northwest Pacific Ocean. At the same time, negative correlation along the East Asia subtropical front (Meiyu) enhance in the period of 1997–2016. The above results and analysis have been supplemented in the revised manuscript. Thanks again for your valuable comments that helped us substantially improve the logic and quality of our manuscript.

Reference:

Wu R, Hu ZZ, Kirtman BP (2003) Evolution of ENSO-related rainfall anomalies in East Asia. J Clim 16:3742–3758.

Feng J, Wang L, Chen W (2014) How does the East Asian summer monsoon behave in the decaying phase of El Niño during different PDO phases? J Clim 27:2682–2698

Song F, Zhou T (2015) The Crucial role of internal variability in modulating the decadal variation of the East Asian Summer Monsoon–ENSO Relationship during the twentieth century. J Clim 28:7093–7107

Lu R, Dong B, Ding H (2006) Impact of the Atlantic Multidecadal Oscillation on the Asian summer monsoon. Geophys Res Lett 33: L24701. https://doi.org/10.1029/2006GL027655

Lin J S, Wu B, Zhou TJ (2016) Is the interdecadal circumglobal teleconnection pattern excited by the Atlantic multidecadal Oscillation? Atmos Ocean Sci Lett 9:451–457.

Wu B, Lin J, Zhou T (2016a) Interdecadal circumglobal teleconnection pattern during boreal summer. Atmos Sci Lett 17:446–452

Wu Q, Yan Y, Chen D (2016b) Seasonal prediction of East Asian monsoon precipitation: skill sensitivity to various climate variabilities. Int J Climatol 36:324–333

Lu, R., 2001: Interannual variability of the summertime North Pacific subtropical high and its relation to atmospheric convection over the Warm Pool. J. Meteor. Soc. Japan, 79, 771-783.

Sun-Seon Lee, Ye-Won Seo, Kyung-Ja Ha, and Jong-Ghap Jhun. Impact of the Western North Pacific Subtropical High on the East Asian Monsoon Precipitation and the Indian Ocean Precipitation in the Boreal Summertime. Asia-Pacific J. Atmos. Sci., 49(2), 171-182, 2013. DOI:10.1007/s13143-013-0018-x

**Specific comments:**

***Point 1***. *The results shown in Fig.1 and 2 are annual mean or winter season?*

**Response:** Here the season we analyzed is summer. The relationship between TP summer temperature and EASM precipitation are mainly performed, and we have specified it in the legend and the text.

*Point 2. Figure 4b. The two rectangular represent north and south parts of East Asia, respectively. However, salient regional difference in summer precipitation can be easily seen in these two domains. This basic feature has also been reported by many literatures. Therefore, it is not reasonable to divide the entire East Asia into only two regions.*

**Response:** Thank you for your comment. We have reviewed the relevant literatures and did find that there is a little different with the basic feature has also been reported by previous literatures (Yu and Zhou 2007; Zhou et al. 2009a; Nigam et al., 2015). The anomalous rainfall pattern is often referred as the South-Flood North-Drought (SFND) pattern, which is performed as a coherent meridional dipole over eastern China, with significant drying in the middle and lower reaches of the Yellow River (where climatological rainfall is less than 4 mm/day) and increasing rainfall across and to the south of the Yangtze River is also evident. Therefore, we have reorganized the sentences and do not quote the SFND pattern. Note that it is slightly difference with the spatial distribution of the multiple regression method of TP-EASM precipitation analyzed latterly. Therefore, combining the spatial distribution of the trend and the multiple regression, we mainly divide it into three parts to analysis, namely the southern part of East Asia, East Asia subtropical front zone, and the Inner Mongolia and Northeast China.

Reference:

Yu, R. C., and T. J. Zhou, 2007: Seasonality and three-dimensional structure of the interdecadal change in East Asian monsoon, J. Climate, 20, 5344-5355.

Zhou, T., D. Gong, J. Li, and B. Li, 2009a: Detecting and understanding the multidecadal variability of the East Asian summer monsoon -Recent progress and state of affairs. Meteor. Zeitschrift, 18, 4, 455-467.

Nigam Sumant, Yongjian Zhao, Alfredo Ruiz-Barradas, Tianjun Zhou, 2015: The South-Flood North-Drought Pattern over Eastern China and the Drying of the Gangetic Plain, 437-359pp (Chapter 22) in: Climate Change: Multidecadal and Beyond, edited by Chih-Pei Chang, Michael Ghil, Mojib Latif, John M. Wallace, 2015 World Scientific Publishing Co.

***Point 3***. *Figure 5. What season for temperature in eastern TP? And it is also strange that the summer precipitation during 1979-2016 regressed on the temperature in eastern TP is almost same with that with liner trend removed. Did the author removes the trend in TP temperature and precipitation simultaneously?*

**Response:** The *temperature* in eastern TP is summer (JJA). Only the trend in TP temperature are removed in previous manuscript. In the revised manuscript, we remove the trend in TP temperature and precipitation simultaneously (Figure 6c, 6d), and the pattern is still same with liner trend unremoved.

[Figure]

Figure 6. Standardized partial regression coefficients between TP temperature and EASM precipitation (GPCP) for (c) 1980-2016, (d) the same as Figure 6c but TP temperature and EASM precipitation are detrended. Black spots indicate statistical significance above the 90% confidence level.

***Point 4.*** *Two Rossby wave trains shown in Fig.8a and b are used to explain the possible mechanism of TP warming effect. At least in the lower panel. i.e., the south branch in the lower troposphere, the wave pattern is hard to identify especially for the anticyclone just to the south of TP.*

**Response:** In the revised manuscript, multiple regression method is applied to exclude these factors (ENSO, NPSH, PDO, and AMO), and results are showed in Figure 7. It can be seen from Figure 7b and Figure 7c that, in 1997-2016 and 1980-2016, one wave-like structure in the tropics moves along the low-level monsoon westerly (the red dotted line in Figure 7). A cyclonic circulation is apparent in the Indochina Peninsula and South China Sea. In the south of TP there is characterized by anticyclonic circulation.

[Figure]

Figure 7. Standardized partial regression coefficients between TP temperature and wind field for (a)1980-1996, (b) 1997-2016, (c) 1980-2016 respectively. (d) the same as Figure 5c but after detrending all data. Vectors are ignored when values less than 0.15, and shading indicate statistical significance above the 90% confidence level.

---

## Author Comment (AC3) · 14 Sep 2019

**Responses to Reviewer #1:**

**Comments and suggestions:**

*This paper mainly addresses three key points: (1) the surface temperature-warming rate of TP area is greater than the rate of global warming; (2) the temperature change is related with elevation; (3) the TP warming is one of the factors which influence the East Asian summer monsoon. Linear regression analysis and Mann-Kendal trend test are adopted. Overall, the statistical methods used in this paper are not rigorous. For instance, the linear regression used in this paper does not provide the assessment of goodness of fit or validation. With this problem, all of the diagnosis which based on the regression are not acceptable. So, I would not recommend this paper unless the authors significantly improve their study.*

**Response:** We appreciate the reviewer's valuable comments that allowed us to improve the manuscript substantially. We have carefully considered these comments and revised the manuscript.

To check whether the trend is reliable or not, Mann-Kendall (M-K) trend test and Theil-Sen estimator are both used. Kolmogorov-Smirnov test is applied to check the statistical significance of normal distribution. The significance of all the regression method and other analysis are determined by the standard two-tailed Student's t test method. This part of the content is described in detail in the method. (P2, L170-185). Please see below for our point-by-point response. The original comments are quoted in Italic.

*Major Comments:*

**Comment 1**. *Lines 134-135: "In the eastern part of the TP, stations at altitudes above 2000 m are selected as representative stations." Why stations above 2000m are selected? Why only eastern part?*

**Response:** Thank you for your valuable comments. Lawrimore et al. (2011) pointed out that regions above 5000 m above sea level are mostly unexplored even if stations installed at such high altitudes would be crucial to fully understand hydro-climatic

processes in the mountains. After quality control, if we choose a site with elevation of 4000 meters or 3000 meters, there are only 9 or 35 stations. To get enough samples, here sites with an elevation of more than 2,000 meters are selected, and the altitude are still higher than the surrounding area, which are also representative. In the previous studies, there were also sites selected with elevation of 2000 meters. Su et al., (2017) chose the gird-averaged results above 2000m in the JRA-55 and MERRA reanalysis datasets to quantitative analysis surface warming amplification over the Tibetan Plateau after the late 1990s using surface energy balance equation. Therefore, we think it is reasonable that stations above 2000m are selected to represent TP.

All situ observation stations with elevations over 2,000 meters are provided by the China Meteorological Science Data Network are indicated in Figure S1. As is showed, a majority of the stations are located in the eastern TP (25°–40°N, 90°–110°E), only 6 stations are located outside the region. In this paper, we only selected concentrated areas to represent the TP to avoid the influence of interpolation on the TP surface temperature. The above results and analysis have been supplemented in the paper. (Figure S1, P2, L158-169)

[Figure]

Figure S1. Situ observation stations with elevations over 2,000 meters of the daily climate dataset (version 3.0) provided by the China Meteorological Science Data Network (The contours represent altitudes of 2000 and 4000m respectively).

Reference:

Su J. Duan A. and Xu H.: Quantitative analysis of surface warming amplification over the Tibetan Plateau after the late 1990s using surface energy balance equation, Atmos. Sci. Let., DOI:

10.1002/asl.732, 2017.

Lawrimore J., Menne M., Gleason B., Williams C., Wuertz D., Vose R., Rennie J.: An overview of the global historical climatology network monthly mean temperature data set, version 3, J Geophys Res 116: D19121. doi:10.1029/2011JD016187, 2011.

**Comment 2:** *Line 139-146: Regression analysis and linear correlation are different, please double check the methodologies and cited literatures.*

**Response:** Thank you for reminding. We have removed this part of the content. Climate-land-atmosphere interact are complex systems under the influence of global warming. In addition to TP temperature, one of the important factor affected annual variability of EASM is the El Niño-Southern Oscillation (ENSO; Wu et al. 2003). On decadal scales, the influence of ENSO on the monsoon can also be strongly modulated by the Pacific Decadal Oscillation (PDO; Feng et al. 2014; Song and Zhou 2015). However, the variability of the tropical Pacific can make clear only part of the complex structure of the monsoon. Recently, remote teleconnections on East Asia from the North Atlantic region have garnered considerable attention. For example, the Atlantic Multidecadal Oscillation (AMO) has been linked to multidecadal changes in atmospheric circulation and precipitation over China (Lu et al. 2006; Lin et al. 2016; Wu et al. 2016a, b). The Western North Pacific Subtropical High (WNPSH) is an also crucial component of the EASM system and significantly influences the precipitation in East Asia (Lee et al., 2013). On the whole, global warming and interdecadal natural variability such as ENSO, NPSH, PDO, and AMO also influence the summer rainfall change in this area. The multiple linear regression is used to explain the relationship between one continuous dependent variable and two or more independent variables, and here it is applied to exclude these factors. The significance of all the regression method are determined by the standard two-tailed Student's t test method. The results are added in the revised manuscript (P1, L102-119; P2, L178-185).

Reference:

Wu R, Hu ZZ, Kirtman BP (2003) Evolution of ENSO-related rainfall anomalies in East Asia. J Clim 16:3742–3758.

Feng J, Wang L, Chen W (2014) How does the East Asian summer monsoon behave in the decaying phase of El Niño during different PDO phases? J Clim 27:2682–2698

Song F, Zhou T (2015) The Crucial role of internal variability in modulating the decadal variation of the East Asian Summer Monsoon–ENSO Relationship during the twentieth century. J Clim 28:7093–7107

Lu R, Dong B, Ding H (2006) Impact of the Atlantic Multidecadal Oscillation on the Asian summer monsoon. Geophys Res Lett 33: L24701. https://doi.org/10.1029/2006GL027655

Lin J S, Wu B, Zhou TJ (2016) Is the interdecadal circumglobal teleconnection
pattern excited by the Atlantic multidecadal Oscillation? Atmos Ocean Sci Lett 9:451–457.

Wu B, Lin J, Zhou T (2016a) Interdecadal circumglobal teleconnection pattern during boreal summer. Atmos Sci Lett 17:446–452

Wu Q, Yan Y, Chen D (2016b) Seasonal prediction of East Asian monsoon precipitation: skill sensitivity to various climate variabilities. Int J Climatol 36:324–333

Lu, R., 2001: Interannual variability of the summertime North Pacific subtropical high and its relation to atmospheric convection over the Warm Pool. J. Meteor. Soc. Japan, 79, 771-783.

Sun-Seon Lee, Ye-Won Seo, Kyung-Ja Ha, and Jong-Ghap Jhun. Impact of the Western North Pacific Subtropical High on the East Asian Monsoon Precipitation and the Indian Ocean Precipitation in the Boreal Summertime. Asia-Pacific J. Atmos. Sci., 49(2), 171-182, 2013. DOI:10.1007/s13143-013-0018-x

**Comment 3:** *Line 157-170: Station based and region based trend analysis are done, together with some discussion on interannual variability. In fact, these results suggest that the study did not clarify which part is the trend, which part is the variability. Because the authors themselves also noticed and mentioned that "Between 1979 and 2004, the interannual variability of temperature on the TP is relatively small (except in 1981), basically remaining between 12.6°C and 14°C." Then before, the authors said "A marked increase is apparent in the past 40 years in the eastern part of the plateau, of which 56 stations show a statistically significant trend at the 95% confidence level (solid red dots). There are 16 stations with a trend exceeding 0.5°C/decade, all distributed over the northeast and southwest sides of the TP, among which Mangya Station (No.51886) even has a trend greater than 1.0°C/decade." These results made me wonder whether the trend analysis is reliable as probably there are abnormal years that affect the trend analysis, especially, trend analysis is not robust when the data is short.*

**Response:** Thank you for your valuable comments. There is interannual variability of temperature on the TP, especially after 2005, but a significant increase is still existed

in the past 37 years. To check whether the trend is reliable or not throughout the study period, Mann-Kendall non-parametric test and Theil-Sen estimator are both used. First, Mann-Kendall method is used to test whether there is a regime shift in TP temperature, and the result is showed in Figure 2. A regime shift is found in 1997, and it is tested statistically significant at the 99% confidence level. Then the time is broken into two periods, 1980-1996 and 1997-2016, to check whether the trend is significant in different period. Both Mann-Kendall test and Theil-Sen are applied to calculating linear trend in 1980-2016, 1980-1996 and 1997-2016 (black line) respectively (Figure 3). Both methods show that the trend of the three periods is tested by a certain level of significance. So, it is believed that the trend analysis here is reliable and robust although there are abnormal years. The results and analysis above have been supplemented in the paper. (Figure 2, P3, L194-204)

[Figure]

Figure 2. (a) Interannual change of summer temperature averaged by the TP (red) and global (blue). Trend significant test are from Mann-Kendall test (blue line) and the Theil-Sen estimator (green line). (Units: °C). (b) Mann-Kendall regime shift test of the temperature of the eastern part of the TP. Both are statistically significant at the 99% confidence level.

[Figure]

Figure S2. Trend analysis by Mann-Kendall test (red line) and the Theil-Sen estimator (blue line) for 1980-2016, 1980-1997 and 1998-2016 (black line) respectively.

**Comment 4**: *Line 186-190: "From the results of the two sets of data (Figure 2a and 2b), it is clear that the temperature of the TP and its surrounding areas decrease with increasing altitude, which is consistent with the variation of tropospheric temperature with height. However, it decreases at a rate of 0.43°C–0.45°C/100 meters, which is lower than the lapse rate (0.6°C/100 meters) in the tropospheric 190 atmospheres".*
*This statement is also based on the linear regression, assessments of goodness of fit shall be added. The same problem also exists in Line 199-201 and Line 222-223 and other places.*

**Response:** Thanks for your suggestions.

(1) In the original manuscript, Mann-Kendall (M-K) trend test is used to check whether the trend is reliable or not. The M-K test is a non-parametric test used to detect the presence of linear or non-linear trends in time series data. So, assessments of goodness of fit are already done, and we so sorry that we should clearly point out in the text. In revised manuscript, Figure 2 of the original is arranged in Figure 3. The

red line in Figure 3 are the linear fitting curve; ** represent statistically significant above the 99% confidence level. The details that have been add as follows:

"To further analyzes elevation-dependent warming on the TP, its surrounding areas (20°–40°N, 90°–110°E; dashed frame in Figure 1) are employed, and the results are shown in Figure 3 (** in the top right corner of the picture represent statistically significant above the 99% confidence level)." (P3, L213-215)

(2) In Line 199-201, assessments of goodness of fit are also already done by using Mann-Kendall (M-K) trend test. The details that have been revised as follows:

"Figure 3c and 3d show the elevation-dependence of the warming rates. Both of them show higher elevation, higher warming rate, and linear trends are statistically significant at the 99% confidence level. "(P3, L226-228)

(3) In Line 222-223, assessments of goodness of fit are also already done. The details that have been revised as follows:

"Kolmogorov-Smirnov test is applied to check the statistical significance of normal distribution. If the computed p-value is greater than the significance level alpha=0.05, one cannot reject the null hypothesis that the sample follows a normal distribution. "(P2, L175-177).

"The statistical significance of climate mean analysis is determined by the standard two-tailed Student's t test method. "(P2, L184-185).

"As shown in Figure 4a and 4b, the normal distribution curve fit of the 2000–6000 meters temperature rate is significantly more concentrated and shifted to the right compared to that of 0–2000meters. The average temperature-change rate increases from 0.26°C/decade to 0.39°C/decade (p<0.001). It can be seen that, with the increase of elevation, the temperature-change rate of the TP and its surrounding areas increases significantly. There is the same phenomenon between the MERRA reanalysis data and the observational data despite difference in value. There are also significant differences between 0-2000 and 2000-4000m layers changes. The average temperature-change rate increases from 0.02°C/decade to 0.07°C/decade (p<0.001). Between 2000-4000 and 4000-6000m, the average temperature-change rate increases from 0.07°C/decade to 0.28°C/decade (p<0.001). " (P3, L244-252).

[Figure]

Figure 3. The (a, b) temperature (units: °C) and (c, d) temperature change rate (units: °C/decade) over the TP and its surrounding areas as a function of altitude: (a, c) observations;(b, d) MERRA2. The red line is the linear fitting curve; **, statistically significant above the 99% confidence level.

[Figure]

Figure 4. Distribution of probability density function of temperature trend (a, b, observational data; c, d, e, MERRA2 data; units: °C/decade) in the TP and its surrounding areas. The histogram represents the samples size, and the curve is a normal distribution fit line. The red line represents the mean of sample. The means and SDs of the normal distribution fit are also added in the upper right corner of the diagram. (a), (b) indicates the altitude ranges of 0–2000 m, and 2000–6000 m, respectively. (c), (d), (e) indicates the altitude ranges of 0–2000 m, 2000–4000 m and 4s000–6000 m, respectively. All graphs are statistically significant at the 95% confidence level by Kolmogorov-Smirnov test except (c). All mean tests are passed by student t test.

**Comment 5**: *Line 207: "The large-scale terrain of the TP has a magnifying effect on the warming rate of warm air, …" I could not follow the authors conclusion. This might be speculation without evidence.*

**Response:** Thanks for your advice, we deleted this sentence "The large-scale terrain of the TP has a magnifying effect on the warming rate of warm air, and the temperature increase in the high-altitude region is higher than that in the low-altitude region." Because we just provided a brief discussion about the temperature of the TP and its surrounding areas decreases with elevation by 0.43–0.45°C/100 meters, which is lower than the standard tropospheric lapse rate (about 0.6°C/-100 meters). This is not a strong conclusion of our work due to without the deeper explore in this manuscript. Thanks for your valuable comment again.

***Comment 6:*** *Line 214-218: Authors divide the altitude into three and two levels based on MERRA reanalysis data and observational data. However, there is no explanation on why they did this.*

**Response:** Thank you for your suggests, we have added a brief discussion. In revised manuscript, Figure 3 of the original is arranged in Figure 4. "As shown in Figure 4, Elevation of stations (grids) in TP between 2000-6000 m are selected to analysis. Simply the stations are average divided into three levels, that is 0-2000m, 2000-4000m, 4000-6000. Considering that there are only few stations above 4000-6000 m, here the stations at the ranges of 2000–4000 and 4000–6000 m are merged together into 2000-6000, and the results are shown in Figure 4." (P3, L238-243)

[Figure]

Figure 4. Distribution of probability density function of temperature trend (a, b, observational data; c, d, e, MERRA2 data; units: °C/decade) in the TP and its surrounding areas. The histogram represents the samples size, and the curve is a normal distribution fit line. The red line represents the mean of sample. The means and SDs of the normal distribution fit are also added in the upper right corner of the diagram. (a), (b) indicates the altitude ranges of 0–2000 m, and 2000–6000 m, respectively. (c), (d), (e) indicates the altitude ranges of 0–2000 m, 2000–4000 m and 4s000–6000 m, respectively. All graphs are statistically significant at the 95% confidence level by Kolmogorov-Smirnov test except (c). All mean tests are passed by student t test.

***Comment 7****: Line 220-224: "As shown in Figure 3a, the normal distribution curve of the 2000–4000 meters temperature rate is significantly more concentrated and shifted to the right than that of 0–2000 meters. The average temperature-change rate*

*increases from 0.26°C/decade to 0.38°C/decade, and the variance reduces from 0.05 to 0.04." There is no statistical significant test, it is hard to believe the conclusion. In addition, authors shall make clear about how did they obtain the normal distribution of temperature.*

**Response:** Thanks for your advice, we have added the statistical significant test. Detailed modifications are as follows:

"Kolmogorov-Smirnov test is applied to check the statistical significance of normal distribution. If the computed p-value is greater than the significance level alpha=0.05, one cannot reject the null hypothesis that the sample follows a normal distribution. "(P2, L175-177).

"The statistical significance of climate mean analysis is determined by the standard two-tailed Student's t test method. "(P2, L184-185).

"As shown in Figure 4a and 4b, the normal distribution curve fit of the 2000–6000 meters temperature rate is significantly more concentrated and shifted to the right compared to that of 0–2000meters. The average temperature-change rate increases from 0.26°C/decade to 0.39°C/decade (p<0.001). It can be seen that, with the increase of elevation, the temperature-change rate of the TP and its surrounding areas increases significantly. There is the same phenomenon between the MERRA reanalysis data and the observational data despite difference in value. There are also significant differences between 0-2000 and 2000-4000m layers changes. The average temperature-change rate increases from 0.02°C/decade to 0.07°C/decade (p<0.001). Between 2000-4000 and 4000-6000m, the average temperature-change rate increases from 0.07°C/decade to 0.28°C/decade (p<0.001). "

Note that, in the revised manuscript, MERRA2 reanalysis data is used to analyze the TP temperature and atmospheric circulation, so the study period was changed to 1980-2016. Therefore, the temperature-change rate has a small change but the overall conclusion remains the same. The results and analysis above have been supplemented in the paper. (P3, L244-252).

[Figure]

t test between (a) and (b):
p < 0.001
t test between (c) and (d):
p < 0.001
t test between (d) and (e):
p < 0.001

Figure 4. Distribution of probability density function of temperature trend (a, b, observational data; c, d, e, MERRA2 data; units: °C/decade) in the TP and its surrounding areas. The histogram represents the samples size, and the curve is a normal distribution fit line. The red line represents the mean of sample. The means and standard deviation of the normal distribution fit are also added in the upper right corner of the diagram. (a), (b) indicates the altitude ranges of 0–2000 m, and 2000–6000 m, respectively. (c), (d), (e) indicates the altitude ranges of 0–2000 m, 2000–4000 m and 4000–6000 m, respectively. All graphs are statistically significant at the 95% confidence level by Kolmogorov-Smirnov test except (c). All mean tests are passed by student t test.

**Comment 8:** *Line 238-240: "In this period, the summer precipitation in China is characterized by the so-called "southern flood–northern drought" spatial distribution (Rectangular area from north to south, respectively present north and south eastern*

*Asia region).*" *The interdecadal variations of summer precipitation over 1979-2016 in China has been investigated in many literatures. Authors should check the literatures and provide precise statement.*

**Response:** Thank you for your valuable comment. We have reviewed the relevant literatures and find that there is a little different with the basic feature of "southern flood–northern drought" that has also been reported by previous literatures (Yu and Zhou 2007; Zhou et al. 2009a; Nigam et al., 2015). The anomalous rainfall pattern is often referred as the South-Flood North-Drought (SFND) pattern, which is performed as a coherent meridional dipole over eastern China, with significant drying in the middle and lower reaches of the Yellow River (where climatological rainfall is less than 4 mm/day) and increasing rainfall across and to the south of the Yangtze River is also evident. Therefore, we have reorganized the sentences and do not quote the SFND pattern. The revised sentences are as follow:

"To verify the reliability of the GPCP precipitation data, the analysis result is compared with the observed site precipitation data. Figure 5 shows the spatial distribution of precipitation change using linear fitting in China (a; observation) and East Asia (b; GPCP) during the summers of 1980–2016. Cross symbols indicate statistical significance above the 95% confidence level. The similar patterns are both displayed in Figure 5a and Figure 5b, which is consistent with the results of previous analyses (Xu et al. 2014; Burke and Stott, 2017; Xu et al., 2018). The precipitation in the south of the Yangtze River, such as South China and Jiangsu and Zhejiang, is generally higher, especially in Jiangsu and Zhejiang provinces. Precipitation in the north of the Yellow River and in Southwest China is generally lower" (P3, L268-275)

Reference:

Yu, R. C., and T. J. Zhou, 2007: Seasonality and three-dimensional structure of the interdecadal change in East Asian monsoon, J. Climate, 20, 5344-5355.

Zhou, T., D. Gong, J. Li, and B. Li, 2009a: Detecting and understanding the multidecadal variability of the East Asian summer monsoon -Recent progress and state of affairs. Meteor. Zeitschrift, 18, 4, 455-467.

Nigam Sumant, Yongjian Zhao, Alfredo Ruiz-Barradas, Tianjun Zhou, 2015: The South-Flood North-Drought Pattern over Eastern China and the Drying of the Gangetic Plain, 437-359pp (Chapter 22) in: Climate Change: Multidecadal and Beyond, edited by Chih-Pei Chang, Michael Ghil, Mojib Latif, John M. Wallace, 2015 World Scientific Publishing Co.

*Comment 9: Line 247-248: "In China, the southern flood–northern drought pattern is also seen with the MERRA data, and is extremely significant. "What is the "extremely significant"? Any test to provide evidence.*

**Response:** Thank you for your suggests. We have reorganized the language as following: "The similarity precipitation distribution pattern between the site data and GPCP data confirm that GPCP precipitation data well reflects the precipitation in China and even in East Asia. On account of well perform of the GPCP data, in the later cause analysis, we will use GPCP precipitation data to find the TP-EASM relationship after TP warming amplification." (P3, L277-280)